# Revealing the room temperature superplasticity in bulk recrystallized molybdenum

Wenshuai Chen[1,2,3,7], Xiyao Li ⓘ [4,7], Shenbao Jin[5], Lunwei Yang[2], Yan Li[2], Xueliang He[2], Wanting Zhang[1,2,3], Yinxing Wu[5], Zhilin Hui[2], Zhimin Yang[1,2,3], Jian Yang[2], Wei Xiao[2,3,6], Gang Sha ⓘ [5] ✉, Jiangwei Wang ⓘ [4] ✉ & Zenglin Zhou ⓘ [1,2,3] ✉

Body-centered cubic refractory metallic materials exhibit excellent high-temperature strength, but often suffer from brittle intergranular fracture due to the recrystallization-induced enrichment of trace elements at grain boundaries (GBs). Here, we report a fully-recrystallized pure molybdenum (Mo) material with room temperature (RT) superplasticity, fabricated by a facile method of powder metallurgy, Y-type hot rolling and annealing. By engineering the ultralow concentration of O at GBs, the inherent GB brittleness of Mo can be largely eliminated, which, in conjunction with high fractions of soft texture and low angle GBs, enables a significant development of ordered dislocation networks and the effective dislocation transmission across low angle GBs. Synergy of these factors greatly suppress the brittle intergranular fracture of Mo, contributing to an enhanced deformability of 108.7% at RT. These findings should have general implication for fabricating a broad class of refractory metals and alloys toward harsh applications.

Molybdenum (Mo)-based metallic materials are widely used in aerospace (turbine), radar communication (traveling wave tube), equipment manufacturing (heating body), nuclear industry (fuel cladding) and other key engineering fields, owing to their excellent mechanical properties and creep resistance at elevated temperatures, strong corrosion resistance, excellent electrical and thermal conductivity, etc[1–4]. However, the poor room-temperature plasticity induced by the intrinsic brittleness (originated from the special electronic configuration of transition metals), and the recrystallization-induced brittleness (i.e. the intergranular brittle fracture induced by the oxygen enrichment at grain boundaries) of Mo and its alloys greatly hinder their wide applications as structural materials under extreme conditions[5,6].

Extensive studies have showed that pure Mo is prone to being subject to full recrystallization (or significant grain coarsening) under thermal or mechanical stimulation, which would greatly reduce the deformability of pure Mo and results in low strength and poor plasticity[2,3]. Numerous alloying strategies have been applied to improve the room-temperature (RT) and high-temperature performances of Mo-based materials in recent years, including solution softening (to reduce the Peierls valley strength of screw dislocation motion by the addition of elements with high number of d-shell electrons, e.g. rhenium)[1,7–11], solution strengthening (to increase the resistance of dislocation motion via lattice distortion) and second phase strengthening (to pin the dislocations and inhibit the

[1]State Key Laboratory of Advanced Materials for Smart Sensing, China GRINM Group Co., Ltd., Beijing 100088, China. [2]GRIMAT Engineering Institute Co., Ltd., Beijing 101407, China. [3]General Research Institute for Nonferrous Metals, Beijing 100088, China. [4]Center of Electron Microscopy, State Key Laboratory of Silicon and Advanced Semiconductor Materials, School of Materials Science and Engineering, Zhejiang University, Hangzhou 310027, China. [5]Herbert Gleiter Institute of Nanoscience, School of Materials Science and Engineering, Nanjing University of Science and Technology, Nanjing 210094, China. [6]State Key Laboratory for Fabrication & Processing of Nonferrous Metals, China GRINM Group Co., Ltd., Beijing 100088, China. [7]These authors contributed equally: Wenshuai Chen, Xiyao Li. ✉e-mail: gang.sha@njust.edu.cn; jiangwei_wang@zju.edu.cn; zhouzenglin@grinm.com

recrystallization)[2,3,12–23]. However, the plastic deformation at high temperature is a diffusion-controlled process rather than a slip-controlled one, which largely depends on the lattice bonding force of the matrix itself. More importantly, Mo-based materials with some active alloying elements (e.g. rare-earth elements) cannot be applied in extreme high-temperature and ultra-high vacuum conditions ($\leq 10^{-8}$ Pa), due to the decomposition of the formed Mo-RE-O composite oxide. Therefore, once the working temperature is above 1500 °C, alloying strategies for Mo-based metallic materials may become significantly ineffective in delaying the recrystallization and maintaining the mechanical properties. It is thus natural to ask how to design high-performance Mo-based materials by tuning the intrinsical thermo-mechanical treatment of Mo. Since the low plasticity and

recrystallization brittleness of Mo at RT mainly originate from its inherent brittleness[24–26] and the high sensitivity to grain boundary (GB) cleavage induced by the GB segregation of inevitable trace elements like O, N and P[5,6,27–31], it needs to, on one hand, alleviate the frequently-observed intergranular fracture by GB engineering, and, on the other hand, enhance the dislocation storage capacity within grain interior. However, it remains technologically challenge to fabricate Mo-based materials with RT superplastic deformability.

Here, we develop a facile strategy to engineer the GB "cleanness" and deformability of Mo-based materials, using a simple combination of powder metallurgy, Y-type hot rolling process and annealing to well control the degree of dynamic recovery. The as-fabricated bulk pure Mo material possesses a stable fine-grain after ultra-high temperature

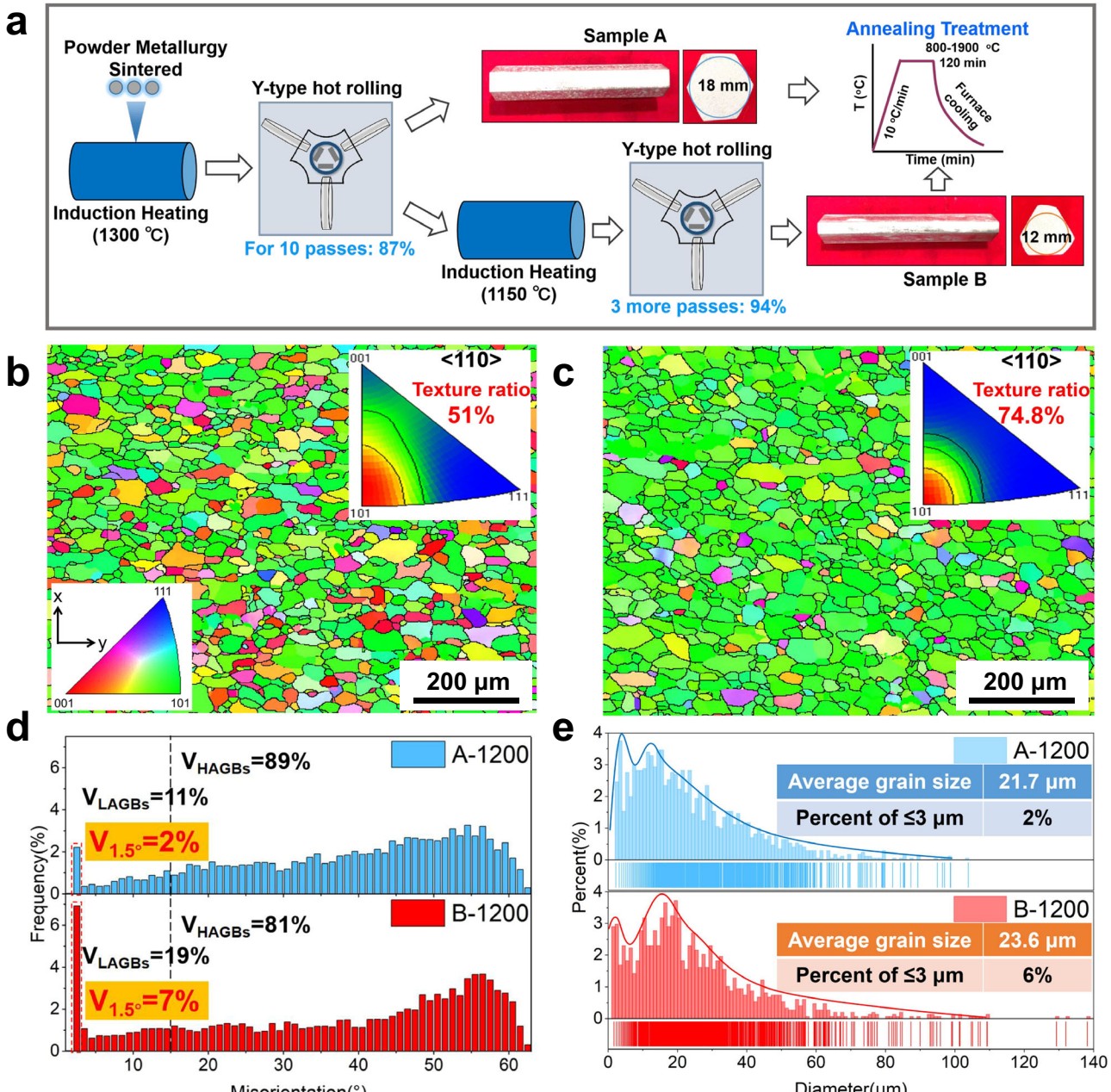

**Fig. 1 | Preparation process and microstructures of Mo bars. a** Two groups of Mo bars A and B were prepared by the powder metallurgy and Y-type hot rolling (with the deformation rate of 87% and 94%, respectively), followed by same annealing process. **b**, **c** After recrystallization, the proportion of <110> //RD texture in sample A-1200 and B-1200 are 51% and 74.8%, respectively. **d** B-1200 sample has a significantly higher proportion of LAGBs, mainly concentrated in 1.5° sub-GB. **e** Grain size distributions in samples A-1200 and B-1200, respectively.

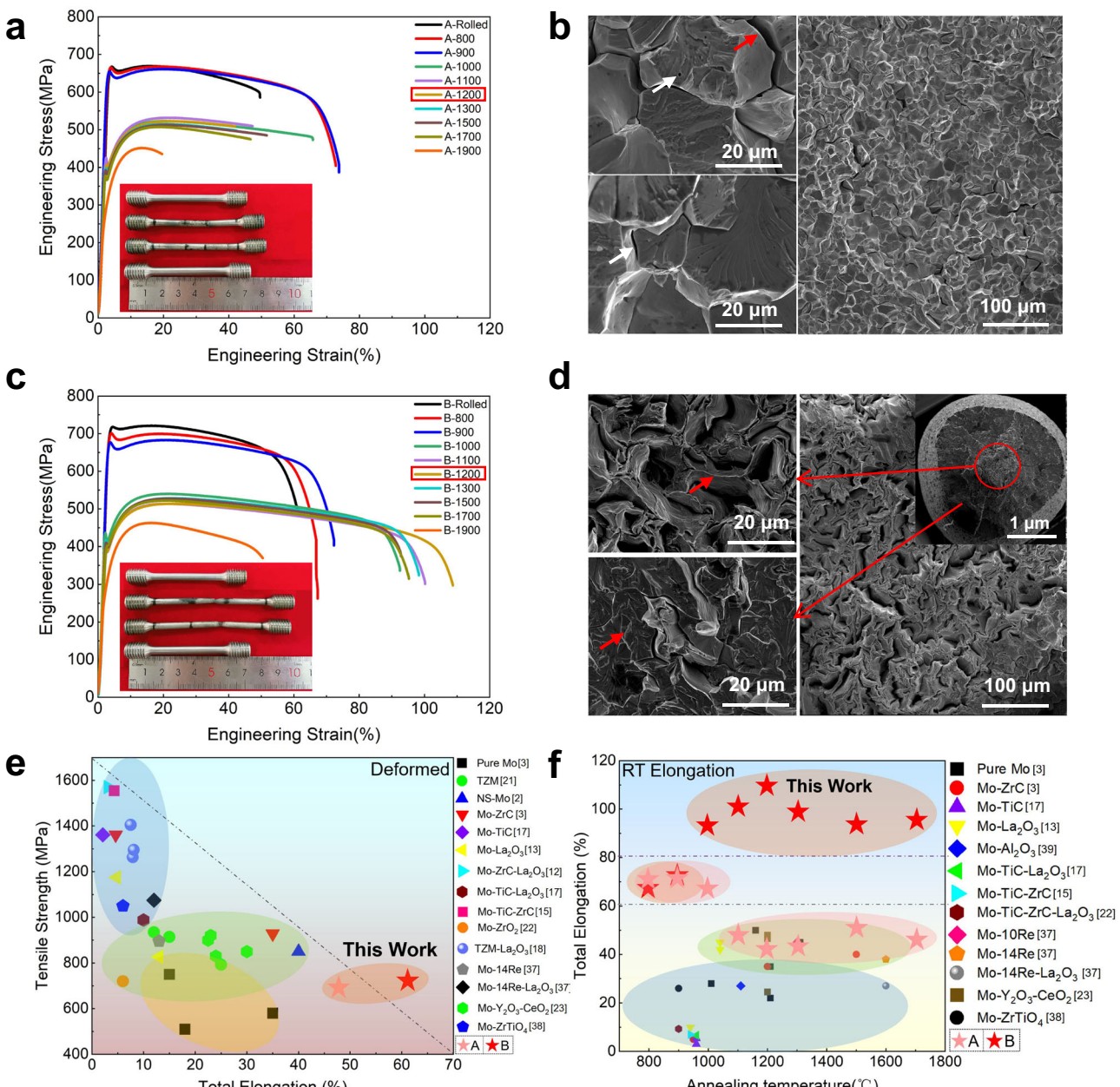

**Fig. 2 | Mechanical properties and tensile fracture morphologies of different samples. a, c** Engineering stress-strain curves and fracture morphologies of samples A and B under different conditions. **b** The A-1200 sample mainly shows an intergranular fracture (see the red arrow), with apparent transgranular cleavage. Certain amounts of micropores with a diameter of <1 μm are observed in the transgranular-fractured grains in A-1200 sample, see the white arrow. **d** The sample

B-1200 exhibits a feature of fibrous tearing in the center and transgranular cleavage in the outer region of the fracture surface, with much smaller cleavage planes. **e** Comparison of tensile strength and total elongation of different as-rolled Mo materials at room-temperature[2,3,12,14,15,17,18,21–23,34–36]. **f** Comparison of total elongation of different annealed Mo materials at room temperature[3,12,14,15,17,18,21–23,34–36].

annealing (1000 ~ 1700 °C) and exhibits an unique superplastic deformability at RT, with a maximum of 108.7% in the 1200 °C annealed samples. Such RT superplasticity of Mo mainly originates from the synergy of proper GB segregation, a high fraction of <110>// RD texture and LAGBs. This low-cost and high-efficiency method would benefit the design and development of strong and ductile refractory metal products for future high-temperature applications.

## Results

### Microstructures of different Mo samples

Different pure Mo samples were fabricated by the powder metallurgy and Y-type hot rolling, see Methods for details. Figure 1a schematically shows the fabrication processes of two typical Mo hexagonal bars,

denoted as A and B, with the reduction rates of 87% and 94% during the Y-type hot rolling, respectively. In the as-rolled samples, both A and B Mo bars possess a slender fibrous structure along the rolling direction (Supplementary Fig. 1), with finer and more uniform fibrous grains formed in sample B due to the higher deformation rate. The ratio of <110>//RD main texture and LAGBs induced in the as-rolled sample B is slightly lower than that of sample A, due to the more sufficient dynamic recovery and recrystallization in the thermomechanical processing. Supplementary Fig. 2 presents the three-dimensional microstructures of samples A and B annealed at different temperatures, where apparent recrystallization occurred as the annealing temperature increasing, with equiaxed grains formed in both samples above 1000 °C. Hardness evolutions in Supplementary Fig. 3 further indicate

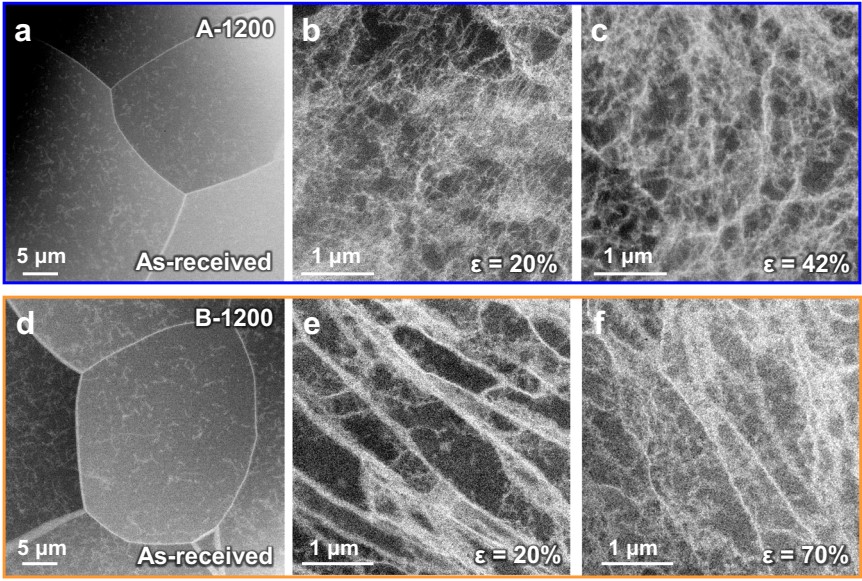

**Fig. 3 | Comparison of intragranular dislocations configurations in samples A-1200 and B-1200 under different tensile strains. a–c** The interaction of dense short dislocations is the main deformation response in sample A-1200.

**d–f** Formation of straight screw dislocations with resultant weaving dislocation networks is the main deformation response in sample B-1200.

the occurrence of full recrystallization after annealing above 1000 °C. Abnormal grain growth occurred in sample A annealed above 1700 °C and sample B annealed above 1900 °C, respectively. Figure 1b, c further demonstrates the microstructures of 1200 °C annealed samples A and B (denoted as A-1200 and B-1200, respectively). After full recrystallization, the ratio of <110> //RD texture in sample A-1200 decreases sharply to 51%, with a slight buildup of <001> //RD texture; while, the ratio of <110> //RD texture in sample B-1200 increases to 74.8%, which can be ascribed to that the recrystallized grains in sample B-1200 inherit the orientation of the previously-deformed grains. Under Y-type hot rolling, a serrated boundary bulges along the GBs of sample B, and dislocations slip, accumulate and rearrange along the convex boundary region, forming a recrystallized core with similar orientation to adjacent deformed grains via a strain-induced GB migration mechanism[32,33]. Therefore, the recrystallized sample B-1200 exhibits a greater degree of <110> //RD texture. Figure 1d further shows the statistical GB structures in samples A-1200 and B-1200. Similarly, both samples are dominated by HAGBs with the fraction of higher than 80%, confirming the full recrystallization; however, with the higher deformation rate of Y-type rolling for sample B, the proportion of LAGBs in fully-recrystallized B-1200, especially 1.5° GBs, increased significantly, with 19% for LAGBs and 7% for 1.5° GBs respectively, in contrast to 11% (LAGBs) and 2% (1.5° GBs) in sample A-1200. Figure 1g demonstrates that the samples A-1200 and B-1200 largely inherit the average grain size of sintered Mo formed by pressing and sintering of fine Mo powders (Supplementary Fig. 4). With the multi-pass three-dimensional thermomechanical treatment, a fine grain size distribution is evidenced after the recrystallization of both samples (Fig. 1e), but the proportion of ultra-fine grains ≤3 μm in the sample B-1200 is higher than sample A-1200.

## Mechanical properties

Figure 2a, c show the tensile engineering stress-strain curves of samples A and B processed under different temperatures (the corresponding true stress-strain curves are presented in Supplementary Fig. 5). Insets in Fig. 2a, c demonstrate the representative macroscale tensile morphologies of A-1200 and B-1200 samples. In the as-rolled state, sample B with a larger hot-rolling deformation can reach a tensile strength of 720 MPa with a total fracture elongation of 61% (with a

uniform elongation of 20%), much higher than 680 MPa and 45% of the as-rolled sample A. Upon low temperature annealing (less than 900 °C), both A and B samples exhibit similar changes of strength reduction and elongation increase (Figs. 2a, c and Supplementary Fig. 6). When annealed above 1000 °C, the tensile strengths of both samples reduce significantly due to the fully-recrystallized structures, which largely keep above 400 MPa for the yield strength and 510 MPa for the fracture strength as the annealing temperature increases; while, the fracture elongation of these two samples exhibit opposite trends in the annealing temperature range of 1000 - 1700 °C. Most notably, after annealing above 1100 °C, the RT elongations of samples A with fully-recrystallized structures are typically less than 50%; in stark contrast, the RT elongations of B samples can further increase to a high level of over 90%, with a maximum of 108.7% obtained after 1200 °C annealing (Some parallel data are shown in the Supplementary Fig. 7), exhibiting a unique RT superplastic behavior in the refractory metals. We further evaluated the tensile behaviors of samples A-1000 and B-1000 under a wide ranges of strain rate (Supplementary Fig. 8), where the sample B-1000 can largely keep its superior mechanical properties, with an uniform elongation of 60% under $3 \times 10^{-4}\,\mathrm{s}^{-1}$. After annealing at 1900 °C, the total elongation of both samples A and B decline markedly due to the annealing-induced abnormal grain coarsening, nevertheless that of sample B-1900 can still reach as high as 50%. Figures 2b, d present the corresponding fracture morphologies of samples A-1200 and B-1200. The sample A-1200 mainly shows an intergranular fracture, with apparent features of transgranular cleavage. Certain amounts of micropores with a diameter of <1 μm are observed in the transgranular-fractured grains in A-1200. In contrast, the sample B-1200 largely exhibits a feature of fibrous tearing at the center of fracture surface, which occurs due to the tensile elongation, rotation, fracture and parallel axial "separation" of <110> -oriented grains. Although some transgranular cleavage also occurs in the outer region of the fracture surface of sample B-1200, the size of these cleavage planes are significantly reduced and the tendency of intergranular GB cracking is greatly alleviated. These fracture morphologies indicate an enhanced plastic deformability of sample B-1200.

The stress-strain curves presented in Fig. 2a, c clearly demonstrate the RT superplastic deformability of sample B-1200. Supplementary Fig. 9 further show that the mechanical properties of as-rolled and

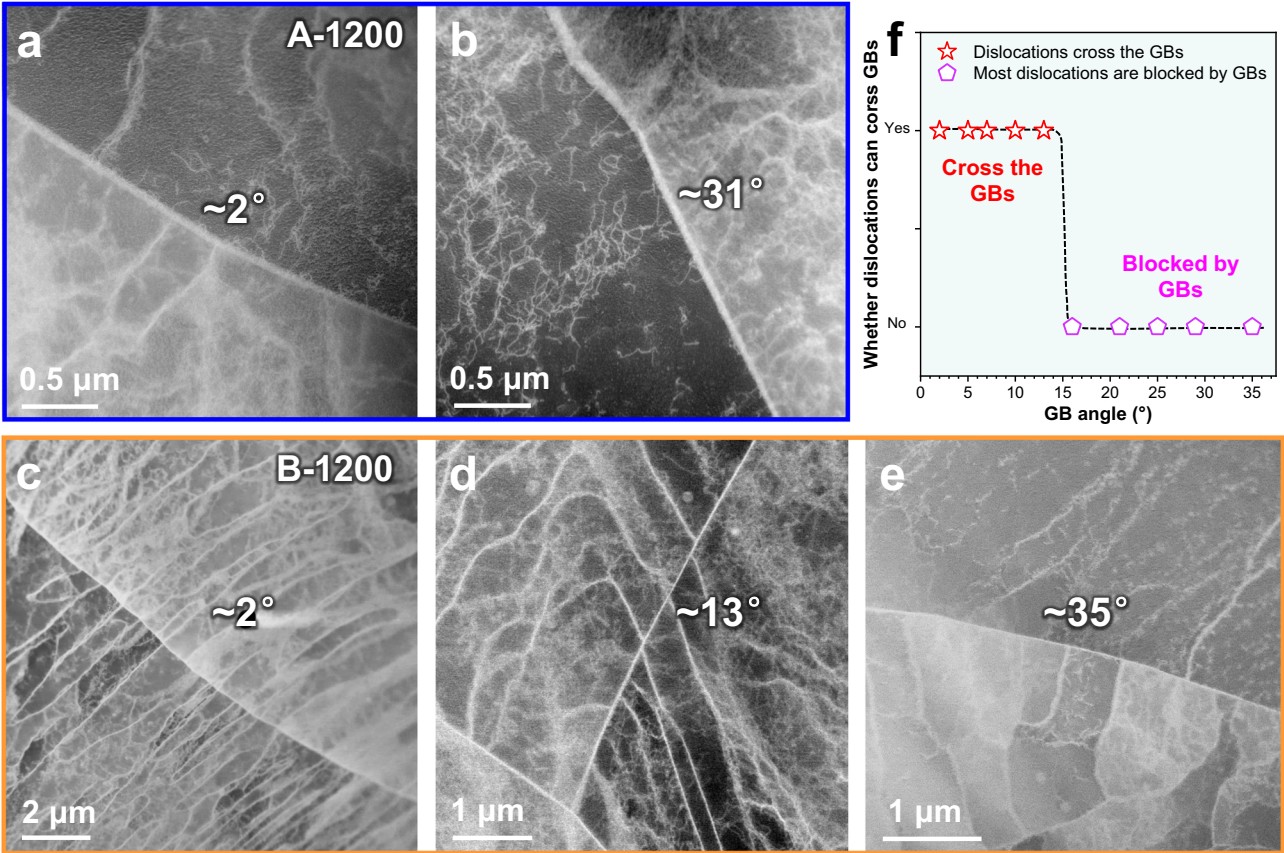

**Fig. 4 | Different GB-dislocation reactions in samples A-1200 and B-1200 at the strain of 20 %. a, b** The GBs act as strong obstacle for the dislocation slip in sample A-1200, no matter LAGBs or HAGBs. **c–f** GB angle-dependent GB-dislocation reaction in sample B-1200. **c–e** The dislocations can cross LAGBs (**c, d**), but are blocked by HAGBs (**e**).

recrystallized Mo bars prepared by the Y-type hot rolling process also exhibit significant advantages, compared with the ones fabricated by the conventional rotary-forging process in the same conditions. Figure 2e summarizes the mechanical properties of different Mo and Mo alloys reported in literatures. Apparently, the as-rolled sample B possesses an elongation notably higher than that of the reported Mo and Mo alloys. After annealing at 1000~1700 °C, a superplastic deformability can be attained in our sample B, with a "jump-up type" improvement compared to the reported plasticity level of the Mo-based materials[2,3,12,14,15,17,18,21–23,34–37] (Fig. 2f).

### Deformation mechanisms in different samples
Considering the great difference in plasticity between samples A-1200 and B-1200, transmission electron microscopy (TEM) observations were conducted to reveal superplastic mechanism, focusing on the intragranular dislocation configurations of these two samples under different plastic strains (Fig. 3). The as-annealed samples A-1200 and B-1200 exhibit similar intragranular dislocation configuration (Fig. 3a, d), with the number densities $\rho_{A-1200}$ and $\rho_{B-1200}$ of $4 \times 10^8 / cm^2$ and $7.9 \times 10^7 / cm^2$ (Supplementary Fig. 10), respectively. However, after a deformation of $\varepsilon = 20\%$, the evolved dislocation configurations in these two samples show strikingly different behaviors. Dense short dislocations are dominant deformation configurations in the sample A-1200 (Fig. 3b); in contrast, long-straight dislocations spread in the grains of deformed sample B-1200 (Fig. 3d), exhibiting a feature of weaving dislocation networks. These weaving dislocation networks act as effective walls to subdivide the grain into small regions, enabling a higher dislocation storage ability, as shown by the dense short dislocations filling the area enclosed (Fig. 3d). In more severely deformed

samples, the tangling of short dislocations proceeds more severely in sample A-1200 ($\varepsilon = 42\%$, Fig. 3c), while the weaving dislocation network is largely kept in sample B-1200 with dense dislocations stored and some dislocation cells formed in between ($\varepsilon = 70\%$, Fig. 3f), contributing to the plastic deformation. Therefore, the interaction of dense short dislocations and the formation of straight screw dislocations with resultant weaving dislocation networks are the main features during the whole deformation process of samples A-1200 and B-1200, respectively. Although the origin of long-straight dislocations remains unclear, such repeated weaving of long-straight dislocations should be responsible for the enhanced plasticity of the bulk recrystallized Mo, by reducing the mean free path for crack propagation[38].

Except for the intragranular dislocation configuration, the GB-dislocation reaction also shows significant differences in samples A-1200 and B-1200. In sample A-1200, the GBs always act as a strong obstacle for dislocation slip, no matter LAGBs or HAGBs (Fig. 4a, b). With such a strong obstacle, dislocation accumulation at GBs can easily induce large deformation incompatibility across the GB, resulting in the GB cracking and thereby intergranular fracture of sample A-1200. In sample B-1200, however, a GB angle-dependent interaction of dislocation with GBs is exhibited, where dislocations can traverse the LAGBs easily but be impeded by the HAGBs. As shown in Fig. 4c, d, the straight dislocations exhibit a significant bending characteristic when they slip across a ~15° GB but not when across a ~2° GB. This indicates that the impedance of GB on dislocation traversing becomes more serious with the increased GB angle. As the GB angle further increases, dislocations largely lose the ability to transmit across the GBs, at a critical angle of ~15° identified based on the statistical observations (Fig. 4f and Supplementary Fig. 11). Given the high fraction of LAGBs

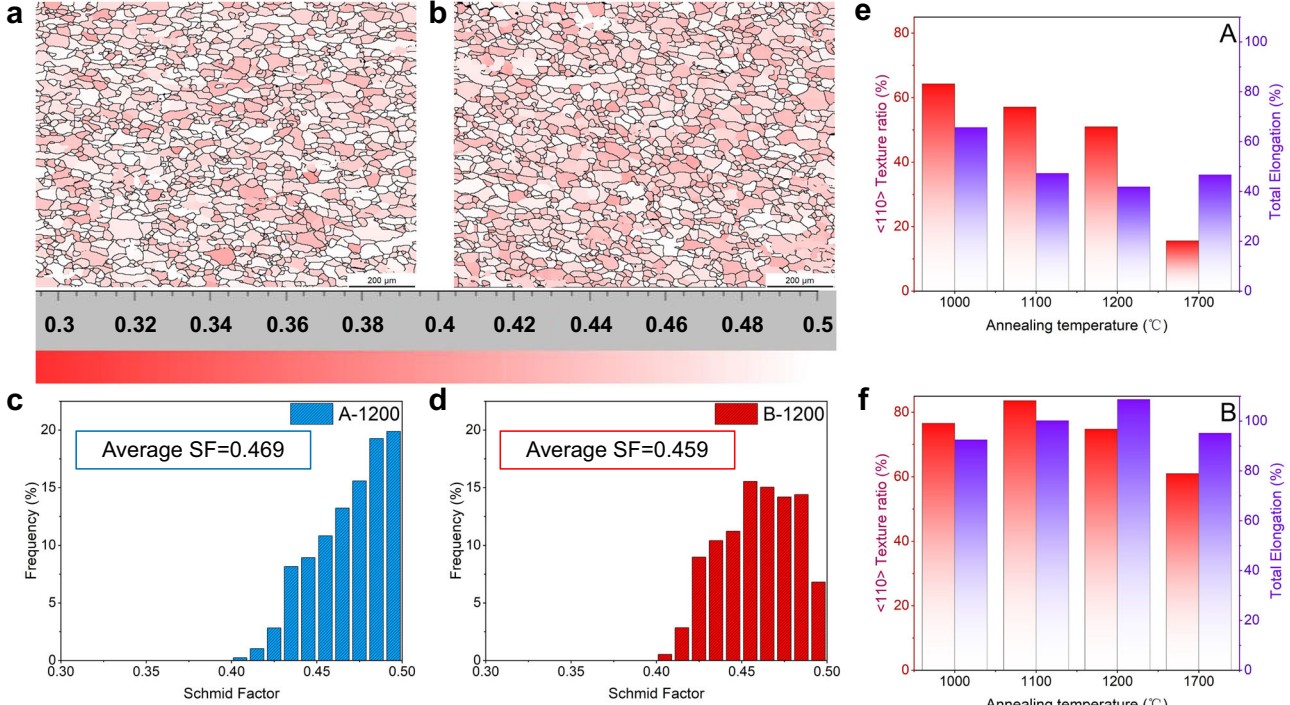

**Fig. 5 | Schmid factor (SF) distributions and the total elongation-<110 >//RD relations for A-1200 and B-1200. a–d** The average SF of A-1200 and B-1200 are 0.469 and 0.459 respectively, which are higher than 0.45. **e–f** The texture components and total elongation of samples A and B annealed at 1000 °C, 1100 °C,

1200 °C and 1700 °C. The ratios of <110 >//RD texture in annealed samples A and B are different (Fig. 1b, c and Supplementary Figs. 12–14), and their respective total elongations did not show a simple linearity with the variation of texture ratio for B samples.

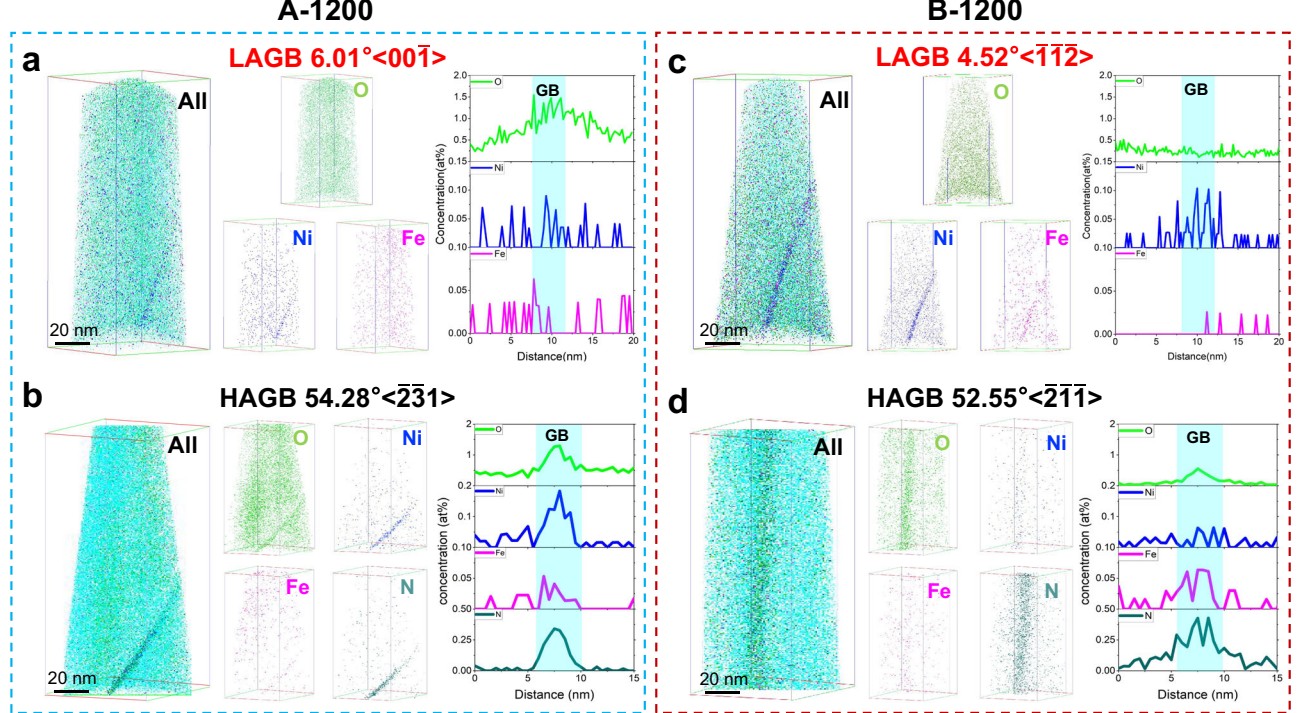

**Fig. 6 | Atom segregations at LAGBs and HAGBs in samples A-1200 and B-1200. a** The O and Ni segregations at a 6°[001̄] LAGB in sample A-1200 are 1.22 at% and 0.04 at%, respectively. **b** The O and Ni segregations at a 54.3°[2̄31] HAGB in sample A-1200 are 1.29 at% and 0.18 at%, respectively. A nitrogen segregation of 0.33 at% also occurs at this HAGB. **c** The O segregation at a 4.5°[1̄1̄2] LAGB in sample B-1200

is as low as 0.19 at%, which is the same as the content of O in the grain interior, and the Ni segregation amount is 0.11 at%. **d** The O segregation at a 52.6°[2̄1̄1̄] HAGB in sample B-1200 is merely 0.55 at%, with the Ni segregation of 0.06 at% and the N segregation of 0.43 at%.

and easy dislocation transmission of LAGBs, dislocation accumulation induced plastic incompatibility at some HAGBs can be easily released by the coordinated deformation of neighboring GBs in sample B-1200 (especially the LAGBs), which can help to eliminate the GB cracking and thereby delay the premature fracture. Note that, in both samples, dislocation transmission across GBs or dislocation emission from GBs can occur occasionally at some locations during the HAGB-dislocation interactions (Fig. 4b and Supplementary Fig. 11d), which may contribute to the plastic deformation to some extent but cannot suppress the GB cracking effectively, as evidenced by the existence of few GB cracking even in sample B-1200 with good ductility in Fig. 2b.

## Role of texture

It is shown that both samples A-1200 and B-1200 are in a soft orientation state with high Schmid factor (SF) by the global SF distribution in Fig. 5, so that intragranular slip systems can be easily activated upon deformation. This can be supported by the dense dislocations upon the deformations of both samples A-1200 and B-1200 (Fig. 3b, e). However, the texture components and total elongation of 1000 °C, 1100 °C, 1200 °C and 1700 °C annealed samples A and B (Fig. 5e, f, respectively) indicate that the <110> //RD texture ratios of samples A and B in annealed states are different (Fig. 1b, c and Supplementary Figs. 12–14), and their respective total elongations did not show a simple linearity with the variation of the texture ratio for samples B. Since the grains of body centered-cubic metallic materials tend to divert to <110> direction during uniaxial tension, the sample B-1200 with a higher proportion of <110> //RD texture should exhibit more uniform intragranular plastic deformation and stronger coordinated deformability between grains[32,33], as supported by Figs. 3 and 4. Thus, we believe that the high ratio of <110> //RD texture is the major cause for the formation and orderly movement of straight screw dislocations with resultant weaving dislocation networks in the grains of sample B-1200. In addition, similar <110> orientation on both sides of GBs also plays a key role in the GB-dislocation interactions and the coordinated deformation between neighboring grains, contributing to the superplasticity of B-series recrystallized Mo.

## Grain boundary chemistry

Technically, pure Mo contains some unavoidable impurities in the range of 100 ~ 1000 ppm, typically, non-metallic elements of O, N and P. In our experiments, trace metal elements also exist in the raw materials, mainly C, O, N, Ni and Fe as detailed in Methods. The contents of O and N in our samples were 32 ~ 34 ppm and 8 ~ 11 ppm, respectively. In particular, the GB segregation of O is expected to significantly weaken the cohesion of GBs in Mo. Given the distinctly different GB distributions in the A-1200 and B-1200 samples, the features of impurity segregation on GBs were further evaluated by the atomic probe tomography (APT). Figure 6 presents the near-atomic scale chemical compositions at the LAGBs and HAGBs of these two pure Mo samples. In sample A-1200, significant enrichment of O element was observed at a 6.01°[00$\bar{1}$]-LAGB (Fig. 6a). The 1D concentration diagram clearly shows that O concentration at the LAGB is higher than that in the grain interior, reaching 1.22 at%. For the quantification of GB segregation content, the Gibbs interface excess (IFE) value was used, which describes the excess number of atoms per unit area (atom/nm$^2$) caused by the presence of an interface. In sample A-1200, the IFE of O at GB reaches 2.79 atom/nm$^2$. In stark contrast, no obvious GB segregation of O element was detected at the 4.52°[$\bar{1}\bar{1}$2]-LAGB of sample B-1200 (Fig. 6c, with a total O concentration of 0.19 at%). Such O concentration at GBs is noteworthily lower than the O segregation level in the sample A-1200 and other pure Mo samples reported in the previous studies[5,6,30,39] (Table 1). Moreover, different from these reported Mo samples, no segregation of N, P and C elements was captured in both A-1200 and B-1200 samples (Table 1); while, trace Ni and Fe were found to segregate at LAGBs, with the levels in A-1200

**Table 1 | Segregation interface excess value of impurity elements in recrystallized pure molybdenum and experimental values in literature**

| Element | Interfacial excess [atoms/nm²] | | | | |
|---|---|---|---|---|---|
| | A-1200 | | B-1200 | | Recrystallized Mo (no Σ-boundaries) |
| | LAGBs | HAGBs | LAGBs | HAGBs | |
| C | — | — | — | — | 0.01–2.2[5] |
| O | 2.79 | 2.95 | — | 1.74 | 0.35–2.0[30,52] |
| N | — | 1.26 | — | 1.48 | 0.03–3.0[6] |
| P | — | — | — | — | 0.15–1.5[6] |
| Ni | 0.04 | 0.47 | 0.14 | 0.07 | No reference |
| Fe | 0.02 | 0.12 | 0.02 | 0.22 | No reference |

The interfacial excess of O at the LAGB of sample A-1200 exceeds the upper limit of the experimental value in the literature[5,6,30,52], and there is no segregation of O at the LAGB of sample B-1200.

sample much lower than that in B-1200 sample (Fig. 6c and Table 1). The peak concentration of Ni at the LAGB of B-1200 sample can reach 0.11 at%, with respect to 0.04 at% at the LAGB of A-1200.

Compared to LAGBs, HAGBs in Mo tend to segregate more solutes which leads to a sharp decrease in GB cohesion and is prone to initiate the intergranular cracks. As exampled in Fig. 6b, d, the 54.28°[$\bar{2}$31] HAGB in sample A-1200 and the 52.55°[$\bar{2}\bar{1}\bar{1}$] HAGB of sample B-1200 tend to segregate more amount of solute elements. The O segregation of sample B-1200 is 0.55 at%, significantly lower than that of A-1200 sample (1.29 at%). However, the Ni segregation at HAGBs in sample A-1200 is 0.18 at%, which is higher than 0.06 at% at HAGBs of sample B-1200. This indicates that Ni is more prone to segregate at the HAGB in sample A-1200, while Ni segregation tendency is the opposite in sample B-1200 with lower Ni content, considering that the Ni content in sample A-1200 is higher than sample B-1200. Significant segregation of N element was observed at the HAGBs of both A-1200 and B-1200, with the values of 0.33 at% and 0.43 at%, respectively. Since the segregation tendency of O and N to GBs in deformed Mo generally depends on the GB angles, GBs with the misorientation angle of <15° in both deformed and recrystallized Mo typically show a lack of N segregation[5]. Our APT results only observed the N segregation at HAGBs, consistent with prvious studies.

Typically, the segregation of Ni and Fe is undoubtedly beneficial for interface bonding, as discussed in previous studies[29,40] and also demonstrated in our Supplementary Discussion; while, the segregation of O and N can significantly reduce the interface bonding, and the weakening effect of O element on GBs is stronger than that of N, which indicates that O plays a more critical role in Mo[5,29,31,39,41–45]. We need to note that the O segregation at LAGB (about zero) and HAGB (about 0.55 at%) in sample B-1200 are much lower than our sample A-1200 and the ones reported in literatures (typically higher than 1 at%)[5], which may greatly help to alleviate the O-induced GB embrittlement in our samples. Moreover importantly, the co-segregation of ultralow O and trace Ni at GBs of sample B-1200 can enable an enhanced ability of bonding rotation and reformation (see the DFT calculation in Supplementary Figs. 15, 16 and Supplementary Table 1), which would benefit the plastic deformation across the GBs. This is validated by the observation of numerous dislocation transmission across the LAGBs in sample B-1200 (see Fig. 4c), in contrast to the limited dislocation transmission in sample A-1200 (see Fig. 4a). Thus, reducing the O concentration should be a practical strategy to enhance the GB fracture strength. For HAGBs, the high fraction of O + N segregation in both sample A-1200 and B-1200 are harmful for the GB cohesion (The O + N segregation and Ni+Fe segregation at the HAGB of A-1200 sample were 1.62 at% and 0.22 at%, respectively; and the O + N segregation and Ni+Fe segregation at the HAGB of B-1200 sample were merely 0.98 at% and 0.12 at%, respectively). In sample A-1200, even though the

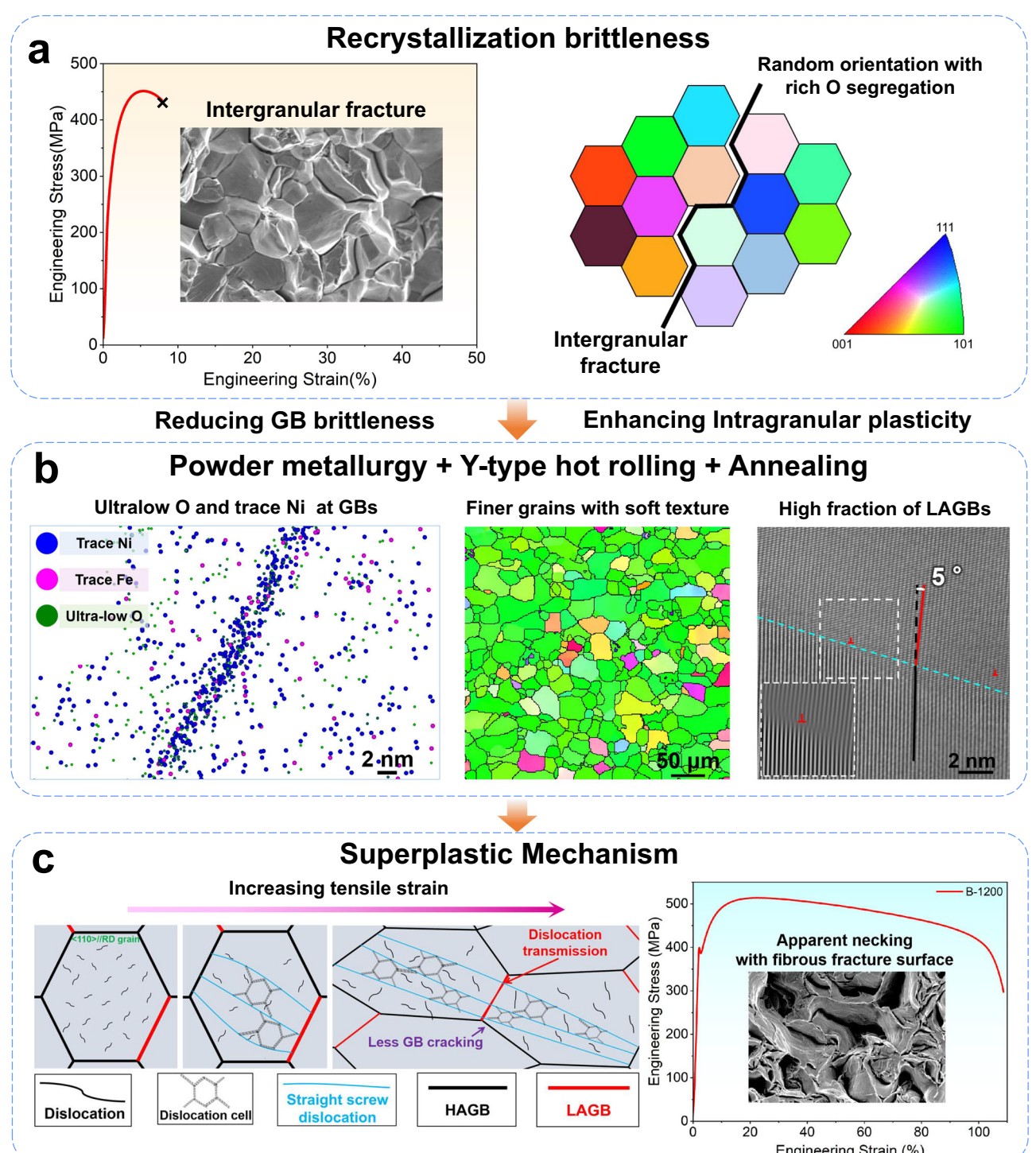

**Fig. 7 | Schematic illustration of superplastic mechanism in sample B-1200.** **a** Recrystallization brittleness caused by randomly oriented coarse grains and oxygen-enriched GBs. **b** Design strategy of Mo materials with superplastic deformability. **c** Superplasticity induced by the unique dislocation behaviors and enhanced GB strength.

HAGB has some Ni+Fe segregation, the much higher fraction of O + N segregation may induce a localized GB decohesion immediately upon localized stress concentration and thereby contribute to a fast GB cracking, as demonstrated by the vast intragranular cracks on the fracture surface of sample A-1200 (Fig. 2b). In sample B-1200, on one hand, the HAGB has a much lower O + N segregation that may help to delay the GB cracking, on the other hand, the higher fraction of LAGBs and vastly-stimulated dislocation transmission mechanism across LAGBs can greatly release the stress accumulated at GBs and in grain interior of sample B-1200, further alleviating the strain localizations at neighboring GBs/grains.

## Discussion

The BCC Mo polycrystalline materials typically exhibits a low ductility and even brittleness at room temperature after recrystallization, mainly induced by the GB cracking due to the interfacial O segregation (see Fig. 7a). To improve the RT plasticity, on the one hand, the frequently-observed intergranular fracture of Mo needs to be

alleviated; on the other hand, the dislocation storage capacity within the grain interior needs to be effectively improved, in order to enhance the intragranular deformability and to slow down the rate of dislocation accumulation at GBs. A finer grain structure with high fraction of LAGB (enabling good dislocation transimission), less O segregation at GBs (enhancing the GB cohesion) and "soft" oriented grains with high SF (improving the intragranular dislocation activities) would benefit the plastic deformaiton of bulk polycrystalline Mo, as schematically shown in Fig. 7b. To obtain above desired structure, we use fine Mo powder as raw material and adopt the powder metallurgy, Y-type hot rolling and annealing process. In contrast to the conventional forging that only subject to a inward radial force, the Y-type hot rolling is more similar to a extrusion+drawing process. With a drawing force along the rolling direction, the <110> orientation of grains in polycrystalline Mo can be managed to rotate to parallel to the deformation direction, while the recrystallization texture mainly depends on the degree of dynamic recovery controlled by hot rolling deformation. As a result, a fine <110> //RD texture can be largely kept after recrytsallization, while a higher fraction of subgrains can be induced by applying a more severe deformation rate. By fully dynamic recovery, a more uniform recrystallized core is formed and competes with each other to grow during annealing, significantly reducing the size of recrystallized grains, and more GBs lead to lower solute element segregation. Thus, a slightly-finer grains with <110> //RD "soft" texture, "cleaner" GBs with reduced O segregation and high fraction of LAGBs can be obtained in our experiments. It also needs to note that the sample B fabricated by our powder metallurgy and Y-type hot rolling method exhibits a good structural stability at high temperature (see Supplementary Fig. 2), which can largely keep the fine equiaxed grains without abnormal grain growth after 1700 °C annealing. It may be attributed to the mutual constraints of the growth of more uniform recrystallization cores, formed through larger thermal deformation[32]. This would benefit the high temperature applications of Mo-based materials.

Slight O (or N) segregation at GBs tends to form Mo-O ion bonds and create a charge density depletion region, which will reduce the GB cohension and give risk to the intergranular fracture of Mo-based materials[5,6,27–31]. Compared with our sample A-1200 and other reported pure Mo[5,6,30,39], the O and N contents at both LAGBs and HAGBs of sample B-1200 are remarkably reduced (O: 0.19 at% for LAGB and 0.55 at% for HAGB, N: zero for LAGB and 0.43 at% for HAGB, see Fig. 6 and Table 1). With the markedly reduced O + N segregation at GBs, the intergranular fracture of Mo can be greatly suppressed. While, the trace segregation of Ni (0.11 at%) can not only help to improve the cohesion of the matrix GBs, but also delay the dislocation pile-up by decreasing the Peierls valley strength of screw dislocation motion[46], further reducing the possibility of intergranular fracture. In contrast, high content of interfacial O segregation in our sample A-1200 and other Mo-based materials not only weaken the cohesion of LAGBs, but also cause a strong resistance for dislocations transmition[47], which can easily induce a fast intergranular fracture.

Upon deformation, the recrystallized finer grains and <110> //RD "soft" texture can help to improve the dislocation activities and dislocation storage ability in the grain interior, enhancing the intragranular plasticity (Fig. 7c). This is shown by the ordered dislocation networks woven by straight spiral dislocations within the <110> //RD soft oriented grains. These dislocation walls can divide the grains into multiple small domains and thus provide additional ability for dislocation storage[38], as evidenced by the formation of some dislocation cells between dislocation walls at 70% strain (see Fig. 3f). The enhanced dislocation storage ability, to some extent, can help to reduce the dislocations pile-up at the GBs compared with the dense and disordered short dislocations overlapping and stacking within the grains of sample A-1200, therefore reducing the stress concentration at GBs. As the deformation proceeds, localized stress concentration at some GBs may reach a certain level. With the high fraction of LAGBs, such stress concentration can be immediately released via the dislocation transmission across LAGBs in the neighbouring regions. The syngery of these factors can help to suppress the intergranular fracture and contribute to the RT superplasticity of sample B-1200.

In conclusion, we develop a bulk pure Mo material with stable fine-grain after ultra-high temperature (1000 ~ 1700 °C) annealing and RT superplasticity via a simple combination of powder metallurgy, Y-type hot rolling and annealing. Such RT superplasticity of Mo-based materials mainly originates from the synergy of proper control of GB segregation, and the introduction of high proportion of <110> //RD texture and LAGBs. The reduced segeration of O and introduction of trace Ni segregation at GBs can greatly suppress the GB brittleness, while the high proportion of <110> //RD texture and LAGBs provides significant capability for the development of ordered dislocation networks, enhanced deformability for dislocation storage, dislocation transmission and stress release, which significantly alleviate the brittle intergranular fracture and enable a RT superplastic deformation in refractory metals. This low-cost and high-efficiency method reported here provides a valuable strategy for the fabrication of Mo-based materials with high structural stability and superior mechanical properties, which should have general implications in a broad class of refractory metals and alloys toward applications under harsh conditions.

## Methods
### Materials preparation
Pure Mo powders (purity ≥ 99.9 wt%, Fisher particle size of $3.0 \pm 0.2$ μm) were used as raw materials, the as-rolled and annealed (unstressed or recrystallized) pure Mo bars were prepared by cold isostatic pressing, hydrogen high temperature sintering, Y-type hot rolling and hydrogen annealing. The Mo powders were loaded into a soft rubber sleeve and cold isostatic pressing was performed at 200 MPa for 10 min to form a round bar compact. It was heated in a hydrogen atmosphere at 1900 °C for 6 h and cooled in a high-temperature furnace. After turning to a diameter of 50 mm, Y-type hot rolling was performed. The Y-rolling temperature was conducted at 1300 °C to obtain hexagonal prismatic Mo bars with diameters of 18 mm (A) and 12 mm (B). The as-rolled A and B samples were heated to 800 ~ 1900 °C at 10 K/min in hydrogen atmosphere for 120 min, then cooled with furnace. The contents of O and N in the samples were measured by oxygen and nitrogen analyzer (LECO, ONH836MC). The contents of other trace elements were measured by glow discharge mass sprumeter (Nu Astrum-ES). The impurity elements in samples A includes C: 20 ppm, O: 32 ppm, N: 11 ppm, Ni: 4.18 ppm, Fe: 7.31 ppm; while, the impurity elements in samples B are C: 20 ppm, O: 34 ppm, N: 8 ppm, Ni: 2.85 ppm, Fe: 5.45 ppm. The matrix contents of Mo in both samples are more than 99.9 wt%. The density of pure Mo samples was measured by Archimedes method, and both samples were nearly fully dense.

### Mechanical testing
The dog-bone-shaped standard room temperature tensile specimens (Supplementary Fig. 17) were prepared by wire cut electrical discharge machining and grinding. The quasi-static uniaxial tensile test was carried out at room temperature with a tensile strain rate of $3 \times 10^{-2}$ s$^{-1}$ by using WDW3100 microcomputer-controlled electronic universal testing machine. The quasi-static uniaxial tensile test was carried out at a tensile strain rate of $3 \times 10^{-4}$ s$^{-1}$ by using YYF-50 slow strain rate stress corrosion testing machine. The original gauge length of the specimens was 25 mm. The tensile strength ($R_m$, MPa), proof strength at plastic extension 0.2% (yield strength, $R_{p0.2}$, MPa), percentage elongation after fracture (A, %) and percentage reduction of area (Z, %) were measured. At least two parallel tensile samples were tested under all experimental conditions to confirm that the data were statistically

valid. During tensile testing at room temperature, a high-resolution strain extensometer was installed over the gauge length to ensure accurate measurement of yield behavior and tensile ductility.

## Structural characterization

The microstructure of the samples along the rolling direction and perpendicular to the rolling direction was observed by metallographic microscope (Zeiss Axiovision 200 MAT). The cut samples were mechanically ground and polished, and then electrolytically polished with 2 wt% NaOH aqueous solution under a voltage of 15 V for 10 ~ 20 seconds. The field emission scanning electron microscope (FEI QUAN FEG 450) equipped with EBSD probe was used for detection. The HKL Channel 5 system was used for data processing to obtain information such as texture component, grain boundary distribution, grain size distribution and Schmid factor distribution. The test step length is 1 μm, the test zone is 1000 μm × 1000 μm, and the acceleration voltage is 15 kV. The fracture morphology of tensile specimens at room temperature was observed by field emission scanning electron microscope (JSM-F100). Thin foils for transmission electron microscopy (TEM) were prepared following the standard procedure, which were mechanically polished down to 70 μm by SiC sandpapers and finally prepared by twin-jet electro-polished until perforation. The electro-polishing was performed using StruersTenuPol-5 at the temperature of 238 K and a voltage of 40 V, within a chemical solution of 130 ml $H_2SO_4$ and 870 ml $CH_3OH$. TEM images were obtained using an FEI Tecnai G2F20 at 200 kV and FEI Titan G2 60-300 at 300 kV. High-angle annular dark-field scanning transmission electron microscopy (HAADF-STEM) imagining was performed on a spherical aberration probe corrected FEI Titan G2 80-200 ChemiSTEM operated at 200 kV. Dislocation densities (ρ) were determined by surface intersection counts[48], following the formula $\rho = 2N/A$, where N is the number of intersection counts of dislocations with both surfaces of TEM foils, A is the corresponding area.

## Elemental composition analysis at atomic scale

The elemental compositions and segregations at grain boundaries were characterized by APT using LEAP 4000X Si. APT samples were prepared in a FIB workstation by annular milling, where the voltage and current were gradually decreased with decreasing specimen diameter. APT data were collected in laser pulsing mode at 50 K with a laser energy of 200 pJ, a pulse rate of 200 kHz and a detection rate of 0.5%, under a high vacuum of around $3 \times 10^{-11}$ mbar. The Cameca integrated visualization and analysis software IVAS 3.8.4 was used for data analysis and three-dimensional atom map reconstruction.

## Calculation of grain boundary strength

The first-principle calculations with density functional theory (DFT) were implemented in the Vienna Ab initio Simulation Package (VASP)[49,50]. The electron-ion interactions were employed by the Generalized Gradient Approximation (GGA) within the framework of Perdew, Burke and Ernzernhof (PBE)[51]. In our simulations, the cutoff energy of plane wave basis set was set as 500 eV, and all the supercell structures were relaxed until the energy and forces acting on any atom were less than $10^{-6}$ eV and 0.01 eV/A, respectively.

## Data availability

The data that support the findings of this study are available from the corresponding author upon request.

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

## Acknowledgements

We would like to thank the financial support from the National Key R&D Projects (2017YFB0306000) and the SASAC Key R&D Projects (2020YQGJ064-012). J.W. acknowledge the financial support from the National Natural Science Foundation of China (52071284) and the Joint Fund of Henan Province Science and Technology R&D Program (225200810058). The authors are grateful to Li Meng and Ning Zhang at the Central Iron and Steel Research Institute for EBSD characterization. We would like to thank Ligen Wang at the General Research Institute for Nonferrous Metals for simulation. We thank Yuan Wu, XiaoYuan Yuan and Huihui Zhu in the University of Science and Technology Beijing for the help with the 3D APT experiments and Qingsong Deng in the Beijing University of Technology for help with the FIB experiments.

## Author contributions

Z.L.Z. designed the project. Z.L.Z., Y.L., W.S.C, X.L.H., W.T.Z and Z.L.H designed the material and processing method, performed mechanical testing and SEM characterizations. X.Y.L, W.S.C, and J.W.W conducted STEM characterizations. S.B.J, W.S.C, Y.X.W. and G.S. carried out APT investigation and data analysis. L.W.Y and W.X conducted the simulation. W.S.C, X.Y.L, L.W.Y, J.W.W. and Z.L.Z. wrote the paper; Z.M.Y. and J.Y helped with data analysis and paper revision. All authors contributed to the discussion.

## Competing interests

The authors declare no competing interests.
