## [Peer Review File · Nature Communications]

Revealing the Room Temperature Superplasticity in Bulk Recrystallized MolybdenumREVIEWER COMMENTS

Reviewer #1 (Remarks to the Author):

The present manuscript reports on the room temperature plasticity of Mo alloys by comparing two Mo alloys mainly differing in their deformation ratio. The results are obtained from a multi-method characterization approach including APT, TEM, EBSD, and mechanical testing and some first principles calculations. The overall work is compelling in its storyline, is written well, and should attract easily an interested readership.

There are some minor comments for the characterization approach, but the major issue is the computational part, which seems in comparison weak and not very helpful:

Major comment:

It is in the spirit of many current works to combine experimental studies with simulations, but this does not always make sense, especially if this information is already available. In the present case, the segregation and embrittlement tendencies of different solutes and impurities are well known and the effect of multiple elements at the GB was also studied before in abundance by multiple authors, and even for the same GB with similar solutes namely C, O, B and Fe instead of O and Ni (Scheiber et al. 2018). Redoing these calculations does not add any value, but raises a lot of issues that have not been detailed in the manuscript. My suggestion is to remove the computational part and refer to available literature due to these questions that are still unanswered:

- a) Details like lattice parameter, cell size, positions for solutes have not been presented.
- b) Convergence study for segregation/embrittlement energies is not present.
- c) How can a general HAGB, i.e. the CSL Sigma 3 GB be representative for segregation states observed at LAGB?
- d) Why was C at the GB computed? It seems to not be present in the APT analysis.
- e) Which specific configurations were studied for the analysis of GB cohesion?
- f) What segregation energies were obtained?
- g) How does the ELF help to understand what the effect of a solute on GB cohesion is, could this be generalized to other GBs, other solutes, other defects? Is there a quantitative measure to be obtained from such an analysis or does one have to depend on visual appearance?

Minor comments:

- 1) Page 6, Line 117: What does Ref. 36 reference to for the sentence in question? The Reference seems to have no connection to Fig 1 or Samples A and B.
- 2) Fig 2a/c: The authors state that parallel tensile tests have been performed. Please supply them with the supplementary information.
- 3) Fig 2e/f: Please also add the values of Sample A to the plots.
- 4) Page 8, Line 158: comparable to the strength of TZM is somewhat misleading as the difference seems to be more than 10%.
- 5) Fig 3 shows different magnifications for A and B in panels b and e, which complicate direct comparison for weaving/non-weaving networks.
- 6) Fig 6a/b: The APT profile across the GB is not very useful with the large scatter (especially for Ni and Fe), rather, IFE values would make much more sense here, not only in the supplementary. Based on the presented profiles and IFE values, the only element significantly contributing to GB cohesion should be O in Sample A, as this content is exceptionally high also in view of available literature data.
- 7) The chemical composition given in Table 2 in the supplementary would be more useful to have in the main manuscript, maybe only including the elements of interest for GB enrichment (Fe, Ni, C, O, N).
- 8) Another work on fracture type of Mo focused on the chemical dependence and observed B segregation although present only in 1-2 ppm in the bulk (K. Leitner et al 2018). Did the authors notice any B content at GB or in the bulk?
- 9) What about the chemistry at HAGBs? It is known that LAGBs attract typically much less solutes than HAGBs due to weaker interaction energies. Also, HAGB are typically the origin of catastrophic failure and the chemistry there should play a much larger role for fracture.

10) For better comparability, dimensions of dog-bone-shaped tensile tests should be visualized in the supplementary, e.g. gauge length, diameter, ...

Point-to-point response to referees' comments

Manuscript Reference Number: NCOMMS-23-15882

Title: Revealing the Room Temperature Superplasticity in Bulk Recrystallized Molybdenum

Authors: Chen et al.

We sincerely thank the referees for their careful review of our manuscript and valuable comments/suggestions on our work. In the following, the review comments are listed in *italic* and our response to each comment is given in blue. The manuscript has been carefully revised accordingly.

Reviewer #1:

The present manuscript reports on the room temperature plasticity of Mo alloys by comparing two Mo alloys mainly differing in their deformation ratio. The results are obtained from a multi-method characterization approach including APT, TEM, EBSD, and mechanical testing and some first principles calculations. The overall work is compelling in its storyline, is written well, and should attract easily an interested readership.

There are some minor comments for the characterization approach, but the major issue is the computational part, which seems in comparison weak and not very helpful:

Response: We thank the referee for recognizing the scientific importance of our work, as well as providing constructive advice for further improving the manuscript. We have revised our manuscript according to these insightful comments, as shown below.

Major comment:

It is in the spirit of many current works to combine experimental studies with simulations, but this does not always make sense, especially if this information is already available. In the present case, the segregation and embrittlement tendencies of different solutes and impurities are well known and the effect of multiple elements at the GB was also studied before in abundance by multiple authors, and even for the same GB with similar solutes namely C, O, B and Fe instead of O and Ni (Scheiber et al. 2018). Redoing these calculations does not add any value, but raises a lot of issues that have not been detailed in the manuscript. My suggestion is to remove the computational part and refer to available literature due to these questions that are still unanswered:

Response: We sincerely appreciate the reviewer for this nice comment and valuable suggestion. The primary focus of our computational work is to verify the effects of different segregation elements, specifically the coupling of O and Ni, on the grain

boundary cohesion and brittleness in a general $\Sigma 3$ GB in Mo alloy. By exploring these specific aspects, our computational study aims to strengthen the overall argument and contribute to a more comprehensive understanding of the research topic.

After careful consideration and thorough review of the relevant literatures, we acknowledge that the computational part of our study has certain limitations, including the possible redundancy in re-analyzing the segregation and embrittlement tendencies that have been extensively investigated by previous authors. To address these concerns, we have relocated the calculational part to the supplementary document and further reinforced our conclusions by incorporating additional references, as follow:

Fig. R1 Simulated structure of Mo $\Sigma 3(111)[110]$ GB and calculation of interface bonding of recrystallized Mo material. **a** Simulated structure of a typical Mo $\Sigma 3(111)[110]$ GB containing interstitial atoms. **b** Relative separation work (percentage) of O, Ni, Fe element segregation and O, Ni co-segregation at the typical Mo $\Sigma 3(111)[110]$ GB, showing that the GB fracture strength decreases significantly with the increase of O concentration, while the co-segregation of Ni with O can help reducing the GB brittleness to some extent. **c** The 2D ELF diagram of O segregation at Mo GB. The color between interstitial O atom and adjacent metal atoms changes from orange (~ 0.8) to blue (~ 0.1), indicating that the outer electrons of metal atoms are captured by O and forming typical ionic bonds; while, the color around the interstitial Ni atoms is close to white ($0.3-0.4$), forming new and stronger Mo-Ni metallic bonds, which is considered to enhance the GB adhesion. The 2D ELF diagram of O and Ni co-segregation at Mo GB shows that the electron localization in the blue region

between Mo-O is weakened, due to the additional free electrons brought by the segregated Ni atoms at the GBs. Therefore, the brittleness of the GBs caused by Mo-O bonding is decreased, which can help to increase its resistance against intergranular fracture.

Once again, we are grateful for the insightful comment provided by the reviewer and appreciate the opportunity to address this concern in our revised version. Please find the replies to the following questions as follows.

a) Details like lattice parameter, cell size, positions for solutes have not been presented.

Response: The relaxed body-centered cubic (BCC) Mo unit cell has a cubic lattice parameter of 3.149 Å. The general Mo $\Sigma 3(110)$ twin grain boundary (GB) models are obtained using a coincidental site lattice (CSL) approach. The fully relaxed Mo $\Sigma 3$ GB consists of 124 atoms, with the dimensions of 8.89 Å \times 7.69 Å \times 34.84 Å, and a vacuum thickness of 7.5 Å along the z-axis. The segregation atoms, e.g. Ni, Fe, O and N, are introduced at the Mo GB as indicated by the blue dotted lines in the structural model in Fig. R1(a). This is the preferred segregation site for them, and has been determined in previously works [1-2].

References:

[1] Janisch R, Ochs T, Merkle A, et al. Structure and stability of grain boundaries in molybdenum with segregated carbon impurities. *MRS Online Proceedings Library (OPL)* 578(1999).

[2] Tran R, Xu Z, Zhou N, et al. Computational study of metallic dopant segregation and embrittlement at molybdenum grain boundaries. *Acta Mater.* **117**, 91-99 (2016).

b) Convergence study for segregation/embrittlement energies is not present.

Response: The segregation energies refer to the energy change associated with preferential accumulation of specific atoms or impurities at GBs within a material [3]. These energies are commonly used to understand and predict the behavior of materials, such as the embrittlement or strengthening of GB in alloys. Tran R et. al. [3] had proposed that the segregation energy difference in GBs and their surface slabs is equivalent to the difference in work of fracture (similar to the separation work) between the doped and undoped GBs. Consequently, we have defined the separation work as $W_{sep} = \frac{E_{FS1} + E_{FS2} - E_{GB}}{S}$, which can be calculated by determining the energy difference between the initial state (intact doped GBs) and the final state (separated doped GBs), and then dividing it by the cross-section area of the GB. The separation work and relative separative work are commonly employed to quantify the resistance or strength of the bond or interaction between atoms or molecules at GBs. These

measurements are frequently employed to assess the bonding fracture and embrittlement characteristics of alloys [4-6].

To evaluate the accuracy and convergence of our simulations, we examined the separation work of the pristine Mo $\Sigma 3$ GB, as well as the relative separation work of the Mo $\Sigma 3$ GB with segregation of one interstitial O atom, which is -5.78 %. This investigation is in line with a previous study [6].

References:

- [3] Tran R, Xu Z, Zhou N, et al. Computational study of metallic dopant segregation and embrittlement at molybdenum grain boundaries. *Acta Mater.* **117**, 91-99 (2016).
- [4] Tahir A M, Janisch R, Hartmaier A. Ab initio calculation of traction separation laws for a grain boundary in molybdenum with segregated C impurities. *Model Simul Mater Sc* **21(7)**, 075005 (2013).
- [5] Lenchuk O, Rohrer J, Albe K. Cohesive strength of zirconia/molybdenum interfaces and grain boundaries in molybdenum: A comparative study. *Acta Mater.* **135**, 150-157 (2017).
- [6] Jing K, Liu R, Xie Z M, et al. Excellent high-temperature strength and ductility of the ZrC nanoparticles dispersed molybdenum. *Acta Mater.* **227**, 117725 (2022).

c) *How can a general HAGB, i.e. the CSL Sigma 3 GB be representative for segregation states observed at LAGB?*

Response: Even though (high-angle grain boundaries) HAGBs and (low-angle grain boundaries) LAGBs have distinct misorientation angles and characteristics, there are still some commonalities that make the $\Sigma 3$ GB a representative model [6] for studying the solute diffusion and segregation states at LAGBs. HAGBs can exhibit similar atomic arrangements, defects, and interfacial properties and energetic considerations in certain regions, leading to similarities in segregation behavior observed in LAGBs, which has been widely studied in literatures [7-10].

Furthermore, we have supplemented our analyses with atom probe tomography (APT) experiments on HAGBs, which also show the less segregation of O (the most important element contributed to the GB embrittlement) at HAGB. This observation is in accordance with the computational findings.

References:

- [6] Jing K, Liu R, Xie Z M, et al. Excellent high-temperature strength and ductility of the ZrC nanoparticles dispersed molybdenum. *Acta Mater.* **227**, 117725 (2022).
- [7] Potin V, Ruterana P, Nouet G, et al. Mosaic growth of GaN on (0001) sapphire: A high-resolution electron microscopy and crystallographic study of threading dislocations from low-angle to high-angle grain boundaries. *Phys. Rev. B* **61(8)**, 5587 (2000).
- [8] Liu Q, Fang L, Xiong Z, et al. The response of dislocations, low angle grain

boundaries and high angle grain boundaries at high strain rates. *Mat Sci Eng A* **822**, 141704 (2021).

[9] Van Swygenhoven H, Spaczer M, Caro A. Role of low and high angle grain boundaries in the deformation mechanism of nanophase Ni: A molecular dynamics simulation study. *Nanostructured materials* **10(5)**, 819-828 (1998).

[10] Fang H, Horstemeyer M F, Baskes M I, et al. Atomistic simulations of Bauschinger effects of metals with high angle and low angle grain boundaries. *Comput Method Appl M* **193(17-20)**, 1789-1802 (2004).

d) *Why was C at the GB computed? It seems to not be present in the APT analysis.*

Response: Thanks for pointing this out. Carbon can be introduced into the samples as contamination during the material processing, which is one reason we calculated C. Besides, the effects of C on GB have been widely investigated in previous studies [6]. To verify the accuracy of our calculations, we have simulated C here, though the C contents in samples A and B are undetectable in our APT experiments. Following the referee's suggestion, we have removed the related data and discussion from the revised manuscript.

Reference:

[6] Jing K, Liu R, Xie Z M, et al. Excellent high-temperature strength and ductility of the ZrC nanoparticles dispersed molybdenum. *Acta Mater.* **227**, 117725 (2022).

e) *Which specific configurations were studied for the analysis of GB cohesion?*

Response: We have chosen the specific configurations of Electron Localization Functional (ELF) to analyze the GB cohesion between Mo and segregated atoms, such as Ni and O in the vicinity of GB.

The ELF is a powerful approach to analyze the distribution of electron density and bonding regions with the Mo GBs, and it can help in understanding the effect of solutes on GBs cohesion by providing insight into the electronic structure and bonding characteristics [11, 12]. By examining the ELF, one can gain knowledge about the nature and strength of chemical bonds between the solute atoms and the atoms in the GB region.

Reference:

[11] ELF: The Electron Localization Function. *Angewandte Chemie International Edition in English* **36(17)**, 1808-1832 (1997).

[12] Wang Z, Wu H, Wu Y, et al. Solving oxygen embrittlement of refractory high-entropy alloy via grain boundary engineering. *Mater. Today* **54**, 83-89 (2022).

f) *What segregation energies were obtained?*

Response: Tran R et. al. [3] had proposed that the segregation energy difference in

GBs and their surface slabs was equivalent to the difference in work of fracture (similar to the separation work) between the doped and undoped GBs. Therefore, we have defined the separation work, which can be calculated by determining the energy difference between the initial state (intact doped GBs) and the final state (separated doped GBs), and then dividing it by the cross-section area of the GB. Separation work and relative separative work are commonly employed to quantify the resistance or strength of the bond or interaction between atoms or molecules at GBs. These measurements are frequently employed to assess the bonding fracture and embrittlement characteristics of alloys [4-6].

According to the definition of segregation energy $E_{seg}^{GB/FS} = (E_{GB/FS+X} - E_{GB/FS}) - (E_{bulk+X} - E_{bulk})$, the table below presents the calculated segregation energies of O, Ni and Fe at the Mo $\Sigma 3$ GB, along with the corresponding data from the reference [13-14]. These values are very close, indicating the reasonable accuracy of our computational work.

	O	Ni	Fe (without spin polarization)
$E_{seg}(X)/eV$	-2.43	-0.348	-0.151
data from	~ -2.50 [13]	~ -0.327 [14]	-0.190 [13]

References:

- [3] Tran R, Xu Z, Zhou N, et al. Computational study of metallic dopant segregation and embrittlement at molybdenum grain boundaries. *Acta Mater.* **117**, 91-99 (2016).
- [4] Tahir A M, Janisch R, Hartmaier A. Ab initio calculation of traction separation laws for a grain boundary in molybdenum with segregated C impurities. *Model Simul Mater Sc* **21(7)**, 075005 (2013).
- [5] Lenchuk O, Rohrer J, Albe K. Cohesive strength of zirconia/molybdenum interfaces and grain boundaries in molybdenum: A comparative study. *Acta Mater.* **135**, 150-157 (2017).
- [6] Jing K, Liu R, Xie Z M, et al. Excellent high-temperature strength and ductility of the ZrC nanoparticles dispersed molybdenum. *Acta Mater.* **227**, 117725 (2022).
- [13] Scheiber D, Romaner L, Pippan R, et al. Impact of solute-solute interactions on grain boundary segregation and cohesion in molybdenum. *Phys Rev Mater* **2(9)**, 093609 (2018).
- [14] Scheiber, Daniel, et al. Ab-initio search for cohesion-enhancing solute elements at grain boundaries in molybdenum and tungsten. *Int. J. Refract. Met. Hard Mater.* **60**, 75-81 (2016).

g) How does the ELF help to understand what the effect of a solute on GB cohesion is, could this be generalized to other GBs, other solutes, other defects? Is there a

quantitative measure to be obtained from such an analysis or does one have to depend on visual appearance?

Response: The ELF is a powerful approach to analyze the distribution of electron density and bonding regions with the Mo GBs, and it could help to understand the effect of solutes on GBs cohesion by providing insight into the electronic structure and bonding characteristics [11, 12]. This analysis can be generalized to other GBs, as well as to other solutes and defects [15-16]. By examining the ELF, one can gain knowledge about the nature and strength of chemical bonds between the solute atoms and the atoms in the GB region.

Except the bonding and electronic structures, ELF can also provide quantitative measures of local electron density. For example, one can calculate the bond strength or bond order from the ELF distribution [17]. This quantitative measure can be useful for comparing the influence of different solutes or defects on GB cohesion [18]. Furthermore, ELF analysis can also be complemented with other techniques such as density functional theory (DFT) calculations or molecular dynamics simulations to obtain a more detailed understanding of solute-GB interaction.

Here, ELF was used to obtain the electron localization of introducing O and Ni atoms around the Mo atom at the GB, in order to determine the bonding strength. It shows that with the segregation of O, a depletion zone of charge density between Mo and O atoms is formed, weakening the Mo-O bonding strength; nevertheless, the ELF values around Ni atoms between the adjacent host metal atoms are very close to that between the adjacent Mo atoms, which would induce new and stronger Mo-Ni metallic bonds in the Ni-segregated GB. The co-segregation of O and Ni at GB of Mo shows that the electron localization in the blue region between Mo-O is weakened, due to the additional free electrons brought by the segregated Ni atoms at the GBs, such that the brittleness of the GBs caused by Mo-O bonding is effectively decreased to some extent, contributing to an enhanced intergranular fracture resistance of Mo materials.

References:

- [11] ELF: The Electron Localization Function. *Angewandte Chemie International Edition in English* **36(17)**, 1808-1832 (1997).
- [12] Wang Z, Wu H, Wu Y, et al. Solving oxygen embrittlement of refractory high-entropy alloy via grain boundary engineering. *Mater. Today* **54**, 83-89 (2022).
- [15] Xiao Z, He L, Bai X M. First principle studies of effects of solute segregation on grain boundary strength in Ni-based alloys. *J Alloy Compd*, **874**, 159795 (2021).
- [16] Lawson J W, Daw M S, Squire T H, et al. Computational Modeling of Grain Boundaries in ZrB₂: Implications for Lattice Thermal Conductivity. *J Am Ceram Soc* **95(12)**, 3971-3978 (2012).
- [17] Chesnut, D. B. Atoms-in-Molecules and Electron Localization Function Study of

the Phosphoryl Bond. *J.Phys.Chem.A* **107**(21), 4307-4313 (2003).

[18] Panek, Jaroslaw Jan, and A. B. Jezierska. "N-oxide Derivatives – Car-Parrinello Molecular Dynamics and ELF Study on Intramolecular Hydrogen Bonds." *J. Phys. Chem. A* **122**, 32 (2018).

Minor comments:

1) Page 6, Line 117: What does Ref. 36 reference to for the sentence in question? The Reference seems to have no connection to Fig 1 or Samples A and B.

Response: We appreciate the reviewer for pointing out this issue. We have decided to delete this expression and Ref. 36.

2) Fig 2a/c: The authors state that parallel tensile tests have been performed. Please supply them with the supplementary information.

Response: Thanks for this suggestion. We have added parallel data for the tensile tests in the supplementary information, as follow:

Fig. R2 Engineering stress-strain curves of different Mo samples. a Engineering stress-strain curves of samples A after rolling and different annealing. **b** Engineering stress-strain curves of samples B after rolling and different annealing. **c-d** Parallel engineering stress-strain curves for samples A and B, respectively.

3) Fig 2e/f: Please also add the values of Sample A to the plots.

Response: We appreciate the reviewer for reminder and have added the data of sample A to Fig. 2e and 2f, as follow:

Fig. R3 Mechanical properties and tensile fracture morphologies of Mo material.
e Comparison of as-rolled room-temperature tensile strength and total elongation at fracture of sample A and B with the reported level^{2,3,12,14,15,17,18,21–23,36–38}. **f** Comparison between the total elongation at fracture at room temperature of sample A and B annealed at different temperatures and the reported level^{3,12,14,15,17,18,21–23,36–38}.

4) Page 8, Line 158: comparable to the strength of TZM is somewhat misleading as the difference seems to be more than 10%.

Response: We appreciate the reviewer for pointing out this issue. This is a comparison of lack of rigor and we have decided to delete this expression, as follow: *Apparently, the as-rolled sample B possesses an elongation notably higher than that of the reported Mo and Mo alloys.*

5) Fig 3 shows different magnifications for A and B in panels b and e, which complicate direct comparison for weaving/non-weaving networks.

Response: We appreciate the reviewer for pointing out this issue and we have unified the magnifications, as follow:

Fig. R4 Comparison of intragranular dislocations configurations in samples of A-1200 and B-1200 under different tensile strains. a-c The interaction of dense short dislocations is the main deformation response in sample A-1200, while **d-f** the formation of straight screw dislocations with resultant weaving dislocation networks is the main deformation response in sample B-1200.

6) *Fig 6a/b: The APT profile across the GB is not very useful with the large scatter (especially for Ni and Fe), rather, IFE values would make much more sense here, not only in the supplementary. Based on the presented profiles and IFE values, the only element significantly contributing to GB cohesion should be O in Sample A, as this content is exceptionally high also in view of available literature data.*

Response: We agree with the referee that the IFE value can better reflect the grain boundary segregation of different impurity elements in A-1200 and B-1200 samples. Following the referee's suggestion, we have inserted the table containing IFE values of Mo and available literature into the main manuscript with some discussions about the contribution of O to GB cohesion.

7) *The chemical composition given in Table 2 in the supplementary would be more useful to have in the main manuscript, maybe only including the elements of interest for GB enrichment (Fe, Ni, C, O, N).*

Response: Thanks for this good suggestion. We have incorporated the related data into Methods section in the main manuscript.

8) *Another work on fracture type of Mo focused on the chemical dependence and observed B segregation although present only in 1-2 ppm in the bulk (K. Leitner et al 2018). Did the authors notice any B content at GB or in the bulk?*

Response: We do aware the effects of different elements on the grain boundary behaviors in Mo. According to the relevant literatures [12,19], the element B was shown to be beneficial for the cohesion of GBs. In our studies, unfortunately, no segregation phenomenon of element B has been detected at both LAGB and HAGB of our bulk Mo materials, which was thus not discussed.

[12] Wang Z, Wu H, Wu Y, et al. Solving oxygen embrittlement of refractory high-entropy alloy via grain boundary engineering. *Mater. Today* **54**, 83-89 (2022).

[19] Leitner (née Babinsky) K., et al. Grain boundary segregation engineering in as-sintered molybdenum for improved ductility. *Scr. Mater.* **156**, 60-63 (2018).

9) *What about the chemistry at HAGBs? It is known that LAGBs attract typically much less solutes than HAGBS due to weaker interaction energies. Also, HAGB are typically the origin of catastrophic failure and the chemistry there should play a much*

larger role for fracture.

Response: We are grateful for the insightful comment provided by the reviewer. Impurity elements tend to segregate at HAGBs, which causes the reduction of GB cohesion and even initiates the intergranular cracks. We have supplemented the GB segregation information of impurity elements at about $54.28^\circ[\bar{2}\bar{3}1]$ HAGB of A-1200 and $52.55^\circ[\bar{2}\bar{1}\bar{1}]$ HAGB of B-1200 samples (Fig. R5). The segregation amounts of oxygen at the HAGB and LAGB in sample A-1200 are 1.2~1.3 at%; in contrast, sample B-1200 possesses a much lower oxygen segregation, which are about 0.55 at% for HAGBs and close to zero for LAGBs. The amount of Ni segregation at the HAGB of sample A-1200 is also higher than that of sample B-1200, indicating that Ni is more prone to segregate at the HAGB in sample A-1200, while Ni segregation tendency is the opposite in sample B-1200 with lower Ni content.

Significant segregation of N element was observed at the HAGBs of both A-1200 and B-1200, and N element also had an adverse effect on the GB cohesion. K. Leitner et.al. [19] have reported that the segregation of O and N to GBs in deformed Mo generally increases with the increase of the GB angle. There is almost no N segregation in $<15^\circ$ GBs of deformed Mo and 15° GB of recrystallized Mo. Based on our APT results, we speculate that the N element may tend to segregate at HAGBs.

We need to note that the oxygen segregation at LAGB (about zero) and HAGB (about 0.55 at%) in sample B-1200 are much lower than the ones reported in literatures (typically higher than 1 at%) [19], which may greatly help to alleviate the O-induced GB embrittlement in our samples. As shown in our simulations, the calculation in Fig. R1(b) for the $\Sigma 3$ (60°) GB have showed that the GBs cohesion can be enhanced by the introduction of Ni and Fe elements. Previous simulation studies also demonstrated similar results for different GBs in Mo, where the segregation of Ni and Fe is undoubtedly beneficial for the interface bonding, while the segregation of O and N can significantly reduce the bonding strength of GBs, and the weakening effect of O element on GBs is stronger than that of N. In our samples, APT characterizations have showed that for LAGBs, Ni+Fe segregation in B-1200 sample is 0.13 at%, which is significantly higher than the segregation of 0.08 at% in A-1200 sample. Fig. 7(b,d) further presented the element segregations at HAGB of A-1200 and B-1200 samples. The O+N segregation and Ni+Fe segregation at the HAGB of A-1200 sample were 1.62 at% and 0.22 at%, respectively; while O+N segregation and Ni+Fe segregation at the HAGB of B-1200 sample were merely 0.98 at% and 0.12 at%, respectively.

For the LAGB, the co-segregation of ultralow O and trace Ni in sample B-1200 can enable an enhanced ability of atom bonding rotation/reformation, which would benefit the plastic deformation across the LAGB. This is validated by the observation of numerous dislocation transmission across the LAGB in sample B-1200 (See Fig.

4c), in contrast to the limited dislocation transmission in sample A-1200 (see Fig.4a). For the HAGBs, the high fraction of O+N segregation should be harmful for the cohesion of HAGBs in Mo. In sample A-1200, even though the HAGB has some Ni+Fe segregation, the much higher fraction of O+N segregation may induce a localized decohesion from the GB immediately upon stress concentration and thereby a fast formation of premature necking via GB cracking, as demonstrated by the vast intragranular cracks on the fracture surface of sample A-1200 (Fig. 2b). In sample B-1200, on one hand, the HAGB has a much lower O+N segregation which may help to delay the GB cracking, on the other hand, the higher fraction of LAGBs and vastly-stimulated dislocation transmission mechanism across LAGBs can greatly release the stress accumulations interior of sample B-1200, further alleviating the strain localizations at neighboring GBs/grains. Moreover, the formation of dislocation cells in the grain interiors of sample B-1200 can improve its dislocation storage ability, which, in conjunction with the coordination deformation between grains (due to the high proportion of small grains), can benefit the grain deformation. All of these factors contribute to the room temperature superplastic deformation of sample B-1200.

Fig. R5 GBs segregation and calculation of interface bonding of recrystallized Mo material. a For A-1200 sample, the O and Ni segregation amount at the LAGB of $6^\circ[00\bar{1}]$ LAGB is 1.22 at% and 0.04 at%, respectively. **b** The O and Ni segregation amount of A-1200 sample at the HAGB of $54.3^\circ[\bar{2}31]$ is 1.29 at% and 0.18 at%, and there is a certain degree of N segregation, with a content of 0.33 at%, respectively. **c** The O segregation amount of B-1200 sample at LAGB of $4.5^\circ[\bar{1}\bar{1}\bar{2}]$ is as low as 0.19 at%, which is the same as the content of O in the grain interior, and the Ni segregation amount is 0.11 at%. **d** The O segregation amount of B-1200 sample at HAGB of $52.6^\circ[\bar{2}\bar{1}\bar{1}]$ is as low as 0.55 at%, and the Ni segregation amount is 0.06 at%, the N segregation amount is 0.43 at%.

References:

[19] Leitner née Babinsky, K, et al. On grain boundary segregation in molybdenum materials. *Mater. Des.* **135**, 204-212(2017).

10) For better comparability, dimensions of dog-bone-shaped tensile tests should be visualized in the supplementary, e.g. gauge length, diameter, ...

Response: Thanks for this suggestion. We have added the sample geometry for tensile test (see Fig. R6) as Supplementary Fig. 16.

Fig. R6 Geometry of the dog-bone-shaped standard tensile specimen.

Reviewer #2:

In this paper, the author reports a fully-recrystallized pure Mo material with room temperature superplasticity. The article provides a systematic and organized work. However, some issues are needed to be addressed:

Response: We thank the referee for the positive comments and valuable suggestions, and we have thoroughly revised our manuscript following the Referee's suggestions.

1. What is the theoretical underpinning for the experimental design method? Did you run trials at various temperatures to find the parameter that performed best and was stated in the article? The text needs to be described in more detail.

2. The RM superplasticity of Mo-base materials mainly originates from the synergy of proper control of GB segregation, the introduction of texture and LAGB. How could you achieve "the proper control"? The article focuses more on experimental analysis and does not provide instructions on how to control. A few sentences need to be carefully thought out and restructured.

Response: We are grateful for the insightful comments of (1) and (2) provided by the reviewer. The comments (1) and (2) are closely related, which are thus answered together below.

The plastic deformation of BCC Mo polycrystalline material is mainly controlled by dislocation slip at room temperature, and typically exhibits a low ductility and even brittleness at room temperature after recrystallization. Such brittleness was often ascribed to the intergranular cracking, which occurs due to the low interface bonding induced by the interfacial oxygen enrichment and the high stress concentration caused by the dislocation accumulation at GBs. To improve the room temperature plasticity, on the one hand, the frequently-observed GB cracking or intergranular fracture of Mo need to be alleviated; on the other hand, the dislocation storage capacity within the grain interior needs to be effectively improved, in order to enhance the intragranular plastic deformability and to slow down the rate of dislocation accumulation at GBs. The high bonding strength of GBs themselves and effective dislocation transmission across GBs can help to reduce the GB cracking via a coordinate deformation between grains. A finer grain structure with high fraction of LAGB (enabling good dislocation transportation), less GB segregation (enhancing the GB cohesion) and "soft" oriented grains with high Schmid factor (improving the intragranular dislocation activities) would benefit the enhanced deformability of bulk polycrystalline Mo.

To obtain above desired structure, here we use fine molybdenum powder as raw material and adopt powder metallurgy, Y-type hot rolling and annealing process to achieve this goal. In experiments, we managed to control the degree of dynamic recovery by optimizing the appropriate amount of hot rolling deformation to obtain the recrystallized cores of proper orientation through hot rolling deformation variables.

Hot rolling process is usually applied in material forming. During the process, the microstructural evolution accompanied with dynamic recovery and recrystallization and an obvious texture could be detected, which is closely related to the change of mechanical properties of the alloys [1-2]. In contrast to the conventional forging process which is only subjected to the inward radial force, the Y-type hot rolling is more similar to the extrusion and drawing process, which is not only subjected to the inward radial force, but also to the drawing force along the rolling direction. After such uniaxial tensile deformation, the $\langle 110 \rangle$ orientation of the grains of the body-centered cubic metal polycrystalline is mostly parallel to the deformation direction, that is, the $\langle 110 \rangle$ fiber texture is formed. The recrystallization texture mainly depends on the degree of dynamic recovery controlled by hot rolling deformation. More sufficient dynamic recovery causes a high degree of dislocation rearrangement, forming a higher proportion of subgrains, triggering a higher probability of forming a recrystallization core by the stress-induced grain boundary migration mechanism. With the fine $\langle 110 \rangle$ //RD fiber texture in the as-rolled sample B, we can easily obtain a slightly fine grains with $\langle 110 \rangle$ //RD “soft” texture, “cleaner” GBs with reduced O segregation (due to the existence of high-density GBs) and high fraction of LAGBs after annealing treatment.

The $\langle 110 \rangle$ //RD fibrous texture component with high Schmid factor (SF) possesses the higher deformability along RD, and the higher volume fraction of the $\langle 110 \rangle$ //RD fiber texture mainly accounts for its higher ductility [1-2]. As a result, the oxygen segregation at GBs of the sample can be significantly reduced by allowing oxygen to dissolve in the grains and make it difficult to reach the GBs quickly. The recrystallized finer grains and $\langle 110 \rangle$ //RD “soft” texture can help to improve the dislocation activities and dislocation storage ability in the grain interior, enhancing the intragranular plasticity. The high fraction of LAGBs can effectively alleviate the stress concentration at GBs and thereby the resultant GB cracking, via the dislocation transmission. All of these contribute to the room temperature superplasticity in sample B-1200.

Thus, with the proper combination of powder metallurgy, Y-type hot rolling and annealing, high density of “cleaner” GBs with ultra-low O (at both LAGBs and HAGBs) and $\langle 110 \rangle$ //RD “soft” texture can be effectively introduced, resulting in an unprecedented room temperature superplasticity in sample B-1200. We have further clarified above information in the revised manuscript.

Reference:

1. Dong, R. et al. Correlation between the mechanical properties and the $\langle 110 \rangle$ texture in a hot-rolled near β titanium alloy. *J. Mater. Sci. Technol.* **97**, 165-168 (2022).
2. Li, J. et al. Texture evolution and the recrystallization behavior in a near β titanium alloy Ti-7333 during the hot-rolling process. *Mater. Charact.* **159**, 109999 (2020).

3. In Figs. 2e and f, please mark Refs next to the ingredients.

Response: We appreciate the reviewer for pointing out this issue and we have marked Refs next to the ingredients.

4. The following specific and normative issues should be modified:

a) The scale line (inward or outward) on the XY axis need to be unified.

b) In Fig. 3, there should be spaces before and after “=”.

c) Figs. 5a and b, the font of the scaleplate are too small.

d) Please check and unify the font type and their size for images.

Response: We appreciate the reviewer for pointing out these issues and apologize for these format problems. We have carefully revised and updated our manuscript, according above suggestions.

5. It is better to provide a schematic diagram to describe the mechanism of the superplasticity in your case.

Response: Thanks for this constructive suggestion. A schematic diagram showing the origin of the room-temperature superplasticity of our Mo samples have been presented in Fig. R7. Molybdenum materials often undergo brittle fracture at grain boundaries after high-temperature annealing, especially after recrystallization caused by randomly oriented coarse grains and oxygen-enriched GBs. We introduce fine grains with soft orientation, “cleaner” GBs with reduced O segregation and trace Ni, and high proportion of LAGBs, by the combination of powder metallurgy, Y-type hot rolling and annealing. During the deformation process of B-1200 sample, ordered dislocation networks woven by straight spiral dislocations are formed within the high content of $\langle 110 \rangle // \text{RD}$ “soft” oriented grains, which divides the grains into multiple small domains and thereby provides additional ability for dislocation storage. When dislocation accumulation reaches a certain level during deformation, the stress concentration at GBs can be immediately released via the dislocation transmission across the LAGBs with strong cohesion (enabled by the nearly-zero O segregation and trace Ni segregation) and resultant coordinated deformation of neighboring grains, given the high proportion of “ductile” LAGBs and “soft” texture in our B-1200 samples. At the same time, the formation of dislocation cells in the grains and the dislocations crossing at the LAGBs significantly alleviate the stress concentration at HAGBs, avoiding the premature cracking from the HAGBs. In contrast, during the tensile process, the short and small dislocations in the grain of A-1200 sample undergo obvious intersection and entanglement, which is easy to form dislocation accumulation at the GBs. This is attributed to the higher proportion of $\langle 001 \rangle // \text{RD}$ oriented grains in the A-1200 sample with poor deformability during the uniaxial

tensile. And the GB segregation of O in our A-1200 sample and other Mo-based materials not only weaken the cohesion of LAGBs, but also cause a strong resistance for dislocations transmission, which can easily induce a fast intergranular fracture. This schematic image (Fig. R7) has been added in the revised manuscript as Fig. 7.

Fig. R7 Schematic illustration of superplastic mechanism in sample B-1200. a Recrystallization brittleness caused by randomly oriented coarse grains and oxygen-enriched GBs. **b** Design strategy of Mo materials with superplastic deformability. **c** Superplasticity induced by the unique dislocation behaviors and enhanced GB strength.

REVIEWER COMMENTS

Reviewer #1 (Remarks to the Author):

The present manuscript reports on the room temperature plasticity of Mo alloys by comparing two Mo alloys mainly differing in their deformation ratio. The results are obtained from a multi-method characterization approach including APT, TEM, EBSD, and mechanical testing and some first principles calculations. The overall work is compelling in its storyline, is written well, and should attract easily an interested readership.

There are some minor comments for the characterization approach, but the major issue is the computational part, which seems in comparison weak and not very helpful:

Major comment:

It is in the spirit of many current works to combine experimental studies with simulations, but this does not always make sense, especially if this information is already available. In the present case, the segregation and embrittlement tendencies of different solutes and impurities are well known and the effect of multiple elements at the GB was also studied before in abundance by multiple authors, and even for the same GB with similar solutes namely C, O, B and Fe instead of O and Ni (Scheiber et al. 2018). Redoing these calculations does not add any value, but raises a lot of issues that have not been detailed in the manuscript. My suggestion is to remove the computational part and refer to available literature due to these questions that are still unanswered:

- a) Details like lattice parameter, cell size, positions for solutes have not been presented.
- b) Convergence study for segregation/embrittlement energies is not present.
- c) How can a general HAGB, i.e. the CSL Sigma 3 GB be representative for segregation states observed at LAGB?
- d) Why was C at the GB computed? It seems to not be present in the APT analysis.
- e) Which specific configurations were studied for the analysis of GB cohesion?
- f) What segregation energies were obtained?
- g) How does the ELF help to understand what the effect of a solute on GB cohesion is, could this be generalized to other GBs, other solutes, other defects? Is there a quantitative measure to be obtained from such an analysis or does one have to depend on visual appearance?

Minor comments:

- 1) Page 6, Line 117: What does Ref. 36 reference to for the sentence in question? The Reference seems to have no connection to Fig 1 or Samples A and B.
- 2) Fig 2a/c: The authors state that parallel tensile tests have been performed. Please supply them with the supplementary information.
- 3) Fig 2e/f: Please also add the values of Sample A to the plots.
- 4) Page 8, Line 158: comparable to the strength of TZM is somewhat misleading as the difference seems to be more than 10%.
- 5) Fig 3 shows different magnifications for A and B in panels b and e, which complicate direct comparison for weaving/non-weaving networks.
- 6) Fig 6a/b: The APT profile across the GB is not very useful with the large scatter (especially for Ni and Fe), rather, IFE values would make much more sense here, not only in the supplementary. Based on the presented profiles and IFE values, the only element significantly contributing to GB cohesion should be O in Sample A, as this content is exceptionally high also in view of available literature data.
- 7) The chemical composition given in Table 2 in the supplementary would be more useful to have in the main manuscript, maybe only including the elements of interest for GB enrichment (Fe, Ni, C, O, N).
- 8) Another work on fracture type of Mo focused on the chemical dependence and observed B segregation although present only in 1-2 ppm in the bulk (K. Leitner et al 2018). Did the authors notice any B content at GB or in the bulk?
- 9) What about the chemistry at HAGBs? It is known that LAGBs attract typically much less solutes than HAGBs due to weaker interaction energies. Also, HAGB are typically the origin of catastrophic failure and the chemistry there should play a much larger role for fracture.

10) For better comparability, dimensions of dog-bone-shaped tensile tests should be visualized in the supplementary, e.g. gauge length, diameter, ...

Point-to-point response to referees' comments

Manuscript Reference Number: NCOMMS-23-15882

Title: Revealing the Room Temperature Superplasticity in Bulk Recrystallized Molybdenum

Authors: Chen et al.

We sincerely thank the referees for their careful review of our manuscript and valuable comments/suggestions on our work. In the following, the review comments are listed in *italic* and our response to each comment is given in blue. The manuscript has been carefully revised accordingly.

Reviewer #1:

The authors have improved their work. I have only some remaining questions/comments:

Response: We appreciate the referee's recognition of our improved work, as well as providing constructive advice for further improving the manuscript. We have revised our manuscript according to these insightful comments, as shown below.

a) Comment regarding the response to point b): The difference in segregation energy to surface and to GB is often termed the strengthening or embrittling energy and originates from the Rice-Wang work on separation of GBs due to solute segregation (and not the work by Tran et al., who have only applied this formalism).

Response: We appreciate the reviewer for reminder and we have changed the original reference of the Rice-Wang work in Supplementary References to replace Tran et al. works.

Reference:

[1] Rice J R, Wang J S. Embrittlement of interfaces by solute segregation. *Mater. Sci. Eng. A* 107, 23-40 (1989).

b) Point e) is still unanswered as the authors do not provide information on the specific GB configuration for computing the effect of solutes on GB cohesion, i.e. at which sites in the GB solutes were placed. Is it the GB central site or situated in a layer farther away from the GB. How was this dealt with in the case of co-segregation with more than one solute atom at the GB? Did the authors select always the energetically most favored configurations or arbitrary ones?

Response: The old configurations may have generated confusion due to its perplexing atoms color and the vague labeling of the coordinate system. Moreover, the article and its attachments have not provided a clear explanation of this intricate configuration. So we completely corrected it, as Supplementary Fig. 16.

As shown in the Fig. R1, the core structures of Mo $\Sigma 3(111)[110]$ GB showcases the remarkable placement of O and Ni solutes within the central region of the GB. In the case of GB co-segregation with O and Ni atoms, the several trial geometries are firstly integrated into the optimized GB+4Ni_{sub} with one O atom which is trace in our experiment added to the atomic vacancy in the central region of the GB, and obtained the most stable GB+4Ni_{sub}+1O_{int} through complete structural optimization.

The Mo GB configurations enumerated in Fig. R1 have merged as the most favored structures within our series of trial models employed in DFT calculations, as based on their comparatively lower energy.

Fig. R1 Geometrical structures (left) and the enlarged core structures (right) of Mo $\Sigma 3(111)[110]$ GB segregation with O and Ni atoms.

c) Based on the response to point f), I assume that the choice was arbitrary, which is an issue as there are sites at the GB where solutes prefer to go and others that are not attractive for segregation. It only makes sense to evaluate solute effects on cohesion for sites where solutes tend to go to. Comparing to literature, it seems as if only the central GB site was considered, which is by far not the site most attractive to segregation for Fe and Ni, and has a different effect on cohesion as well. If simulation part is kept, the correct positions of segregating elements should be evaluated, visualized, and taken as starting basis for computing effect on cohesion.

Response: We possessed an understanding of the distinctive segregation behavior occurring within various types of Mo GBs. Fe and Ni atoms are more likely to stabilize in the non-central region of GBs (near the GBs region) of the Mo $\Sigma 5$ twist and tilt GBs [2]. But we choose the $\Sigma 3(110)$ Mo GB which is quite classic HAGBs for Mo. So we found some papers to clarify that the solutes (Fe, Ni, O and N) present in the same $\Sigma 3$ Mo GB center is much more stable:

Fig. R2 Panel (a) shows the GB used for the segregation calculations with numbered GB sites (center), numbered free surface sites (left) and the bulk site (B), while panel (b) contains the geometric variant observed for some segregating elements [3].

Fig. R3 FS segregation energies $E_{seg,FS}$ and GB segregation energies $E_{seg,GB}$ for three sites and the resulting strength of embrittlement SE for Mo. The symbols for the strength of embrittlement (rectangle, triangle or circle) correspond to the symbol of the site with the strongest GB segregation energy [3].

The original text as presented in the referenced [3] is “The obtained GB structure is shown in panel (b) of Fig. R2 and is obtained when placing the sites 2 and 3 in panel (a) of Fig. R2 in line. In Mo, this restructuring of the GB took place for the solutes Mn, Fe, Co, Ni, Cu, Ru, Rh, Pd, Os, Ir and Pt when positioning them at site 2 and for Ni, Cu, Y, Rh, Pd, Ag, Ir, Pt and Au when positioning them at site 3.” [3] It means that Fe and Ni atoms placed near the GB central region of the $\Sigma 3$ GB in Mo will return to the GB central region after structural optimization, and the same applies to the energy calculation results, as shown in Fig. R3.

Fig. R4 (a) The typical sites selected for calculating the segregation energies. O_{bulk} , T_{bulk} refer to the bulk octahedral and tetrahedral interstitial sites far from GB. $O_{\Sigma 3}$, $T_{\Sigma 3}$ refer to the octahedral and tetrahedral interstitial sites near GB. $G_{1\Sigma 3}$, $G_{2\Sigma 3}$ are two open sites right at the GB. The segregation energies of (b) H, (c) O, (d) N and (e) C at these typical sites as a function of the distance to the $\Sigma 3$ GB plane in the [111] direction [4].

In comparison to the bulk site, the segregation energy becomes smaller near the $\Sigma 3$ GB, indicating the tendency of GB segregation (Fig.R4). The most favorable trapping sites is $G_{\Sigma 3}$ for all the light elements. Among all the light elements, the O has the lowest segregation energy, indicating that it should locate more preferentially at the $\Sigma 3$ GB [4].

References:

[2] Tran, R. et al. Computational study of metallic dopant segregation and

embrittlement at molybdenum grain boundaries. *Acta Mater.* **117**, 91-99 (2016).

[3] Scheiber, D., Pippan, R., Puschnig, P., et al. Ab-initio search for cohesion-enhancing solute elements at grain boundaries in molybdenum and tungsten. *Int. J. Refract. Met. Hard Mater.* **60**, 75-81 (2016).

[4] Ma H, Ding X, Zhang L, et al. Segregation of interstitial light elements at grain boundaries in molybdenum. *Mater. Today Commun.* **25(4)**, 101388 (2020).

d) Regarding point g), I urge the authors to employ methods that can be quantified and are not based on visual inspection of colorful images. As they mention, there are methods and tools available for that, e.g. the bond order analysis or (I)-COHP.

Response: Thanks for this constructive suggestion. We employ the bond order to quantified the GB cohesion with the segregation of O and Ni.

Generally, a higher bond order signifies a stronger chemical bond [5,6]. We employed bond order of the core structure at GB center in Fig. R1 and Table R1, which quantitatively characterize the bond strength of Mo GB with the segregation of O and Ni. The bond order of Mo-Mo in the Mo bulk and the GB center are 0.43~0.52 and 0.34~0.38, respectively. As depicted in Fig. R1(b), the bond order of the central Mo atom (No. 1) and the Ni atoms (No. 11, 12, 13) is 0.471, which are remarkably surpassed that of the upper (No. 6, 7, 8) and lower (No. 2, 3, 4) neighboring Mo atoms with band order of 0.350 to 0.389.

The introduction of O atom (No. 13) in both GB+1O_{int} and GB+4Ni_{sub}+1O_{int} generates a formidable bond with the central Mo atom with a respective bond order of 0.432 and 0.498 portrayed in Fig. R1(a) and (c). Meanwhile, a noteworthy shift in the bond strength occurs between the center Mo and the adjacent Mo atoms (No. 4, 8) with the bond order diminished to 0.172/0.203 in GB+1O_{int}. Similar change occurred in the GB+4Ni_{sub}+1O_{int} case. This indicates that the conclusion of the band order is consistent with the ELF calculation.

References:

[5] Manz, T. A. Introducing DDEC6 atomic population analysis: part 3. Comprehensive method to compute bond orders. *RSC Adv.* **7**, 45552-45581 (2017).

[6] Song, J. et al. Thermal instability originating from the interface between organic-inorganic hybrid perovskites and oxide electron transport layers. *Energy Environ. Sci.* **15**, 4836-4849 (2022).

Table. R1 The bond order for GB cohesion with the segregation of O and Ni.

Bond		Bond order		
		GB+1O_{int}	GB+4Ni_{sub}	GB+4Ni_{sub} +1O_{int}
Mo-Mo	1-2	0.337	0.392	0.333
	1-3	0.317	0.392	0.198
	1-4	0.173	0.388	0.403
	1-5	0.191	0.225	0.190
	1-6	0.323	0.350	0.350
	1-7	0.312	0.350	0.270
	1-8	0.203	0.365	0.342
	1-9	0.189	0.202	0.208
	1-10	—	0.471	0.213
	Mo-Ni	1-11	—	0.471
1-12		—	0.471	0.438
Mo-O	1-13	0.498	—	0.432

REVIEWER COMMENTS

Reviewer #1 (Remarks to the Author):

The present manuscript reports on the room temperature plasticity of Mo alloys by comparing two Mo alloys mainly differing in their deformation ratio. The results are obtained from a multi-method characterization approach including APT, TEM, EBSD, and mechanical testing and some first principles calculations. The overall work is compelling in its storyline, is written well, and should attract easily an interested readership.

There are some minor comments for the characterization approach, but the major issue is the computational part, which seems in comparison weak and not very helpful:

Major comment:

It is in the spirit of many current works to combine experimental studies with simulations, but this does not always make sense, especially if this information is already available. In the present case, the segregation and embrittlement tendencies of different solutes and impurities are well known and the effect of multiple elements at the GB was also studied before in abundance by multiple authors, and even for the same GB with similar solutes namely C, O, B and Fe instead of O and Ni (Scheiber et al. 2018). Redoing these calculations does not add any value, but raises a lot of issues that have not been detailed in the manuscript. My suggestion is to remove the computational part and refer to available literature due to these questions that are still unanswered:

- a) Details like lattice parameter, cell size, positions for solutes have not been presented.
- b) Convergence study for segregation/embrittlement energies is not present.
- c) How can a general HAGB, i.e. the CSL Sigma 3 GB be representative for segregation states observed at LAGB?
- d) Why was C at the GB computed? It seems to not be present in the APT analysis.
- e) Which specific configurations were studied for the analysis of GB cohesion?
- f) What segregation energies were obtained?
- g) How does the ELF help to understand what the effect of a solute on GB cohesion is, could this be generalized to other GBs, other solutes, other defects? Is there a quantitative measure to be obtained from such an analysis or does one have to depend on visual appearance?

Minor comments:

- 1) Page 6, Line 117: What does Ref. 36 reference to for the sentence in question? The Reference seems to have no connection to Fig 1 or Samples A and B.
- 2) Fig 2a/c: The authors state that parallel tensile tests have been performed. Please supply them with the supplementary information.
- 3) Fig 2e/f: Please also add the values of Sample A to the plots.
- 4) Page 8, Line 158: comparable to the strength of TZM is somewhat misleading as the difference seems to be more than 10%.
- 5) Fig 3 shows different magnifications for A and B in panels b and e, which complicate direct comparison for weaving/non-weaving networks.
- 6) Fig 6a/b: The APT profile across the GB is not very useful with the large scatter (especially for Ni and Fe), rather, IFE values would make much more sense here, not only in the supplementary. Based on the presented profiles and IFE values, the only element significantly contributing to GB cohesion should be O in Sample A, as this content is exceptionally high also in view of available literature data.
- 7) The chemical composition given in Table 2 in the supplementary would be more useful to have in the main manuscript, maybe only including the elements of interest for GB enrichment (Fe, Ni, C, O, N).
- 8) Another work on fracture type of Mo focused on the chemical dependence and observed B segregation although present only in 1-2 ppm in the bulk (K. Leitner et al 2018). Did the authors notice any B content at GB or in the bulk?
- 9) What about the chemistry at HAGBs? It is known that LAGBs attract typically much less solutes than HAGBs due to weaker interaction energies. Also, HAGB are typically the origin of catastrophic failure and the chemistry there should play a much larger role for fracture.

10) For better comparability, dimensions of dog-bone-shaped tensile tests should be visualized in the supplementary, e.g. gauge length, diameter, ...

Point-to-point response to referees' comments

Manuscript Reference Number: NCOMMS-23-15882

Title: Revealing the Room Temperature Superplasticity in Bulk Recrystallized Molybdenum

Authors: Chen et al.

We sincerely thank the referees for their careful review of our manuscript and valuable comments/suggestions on our work. In the following, the review comments are listed in *italic* and our response to each comment is given in blue. The manuscript has been carefully revised accordingly.

Reviewer #1:

Comments a), b) and d) have been addressed satisfactorily, but I still take issue with the response to comment c):

Response: We appreciate the referee's recognition of our improved work, as well as providing constructive advice for further improving the manuscript. We have revised our manuscript according to these insightful comments, as shown below.

a) As mentioned in Ref [3] (Scheiber et al), Ni placed at site 2 of the GB leads to a reconstruction moving the site to the center of the GB, however, this is not the same as taking immediately site 1 as can be inferred from Fig R3. Whereas Ni at site 2 exhibits a segregation tendency of -2 eV, at site 1 this is only about 0.4 eV, i.e. lower by a factor of 5 and by far not the most stable position. Fe even has a segregation energy of 0 at site 1 compared to -2 eV at site 2! So it makes an immense difference what site is chosen.

Therefore, I still ask the authors to recompute segregation of Ni and Fe at the energetically preferred sites or refer to existing and more accurate modelling treatments.

Response: We greatly value the concerns raised by the Reviewer #1 and have conducted a thorough reassessment and verification of our previous incomplete considerations regarding GB element co-segregation. Based on the complex element segregation environment surrounding the Mo GB, we have taken into consideration some typical substitutional and interstitial sites, such as the atom layers 0, 1, 2, 1' and 2' (Fig. R1) for Ni, Fe, and O segregation in our precomputation. The corresponding free energies are listed in Table R1 and Table R2, respectively.

Fig. R1. The 0, 1, 2, 1', and 2' segregated atom layers for solutes surrounding the Mo GB zone.

Table R1. The free energy of Ni, Fe and O atoms segregate at atom layers 0, 1, 2, 1', and 2' around Mo GB zone.

Solute	Free Energy (eV)				
	atom layer 0	atom layer 1	atom layer 2	atom layer 1'	atom layer 2'
1Ni _{sub}	-1315.49	-1316.36	-1315.27	-1316.45	-1315.27
1Fe _{sub}	-1317.78	-1319.06	-1317.75	-1319.18	-1317.77
4Ni _{sub}	-1303.25	-1303.44	-1302.84	-	-
1O _{int}	-1328.74	-1327.97	-	-	-

Table R2. The free energy of Ni and O atoms co-segregate at atom layers 0 and 1 around Mo GB zone.

Solute	Free Energy (eV)	
	atom layer 0	atom layer 1
$1\text{Ni}_{\text{sub}}+1\text{O}_{\text{int}}$	-1322.97	-1322.59
$4\text{Ni}_{\text{sub}}+1\text{O}_{\text{int}}$	-1309.48	-1309.14

According to the information provided in Table R1, it appears that the energetically stable atom layers for Ni or Fe atoms segregated at the Mo GB zone are 1 ($\approx 1'$) $> 0 > 2$ ($\approx 2'$). For the O atom, the atom layer 0 is more stable energetically than the atom layer 1. It means that the energetically stable sites for O atoms segregated at the Mo GB zone are central atom layers. For the Ni and Fe atoms, the energetically stable sites are sub-central atom layers. These conclusions align with the previous views of Reviewer #1.

Additionally, it is noted that with the segregation of Ni atoms increases, the energy barrier for Mo GB from atom layer 1 to 0 decreases, with a value of only 0.19 eV.

Moreover, as shown in Table R2, when interstitial O atom and substitutional Ni atoms are co-segregated at atom layer 0 (center) and atom layer 1 (sub-center) of Mo GB, the free energy of co-segregation at the central layer is 0.34 eV lower than that at the sub-central layer. It indicates that co-segregation at the central layer is more stable.

Therefore, we believe that it is reasonable to consider atom layer 0 as the position for the co-segregation of Ni and O in the Mo GB, and this can serve the purpose of explaining the experimental observations in the article.

Thank you for elaborating on the information, and we appreciate your insightful contribution.

REVIEWERS' COMMENTS

Reviewer #1 (Remarks to the Author):

The authors have addressed my concerns in sufficient detail.